# Refraction of the Two-Photon Multimode Field via a Three-Level Atom

**DOI:** 10.3390/e27010071

**Published:** 2025-01-15

**Authors:** Trever Harborth, Yuri Rostovtsev

**Affiliations:** Center for Nonlinear Sciences and Department of Physics, University of North Texas, Denton, TX 76203, USA; treverHarborth@my.unt.edu

**Keywords:** quantum optics, quantum information, quantum communications, quantum cavity electrodynamics, three-level atom

## Abstract

Classically, the refractive index of a medium is due to a response on said medium from an electromagnetic field. It has been shown that a single two-level atom interacting with a single photon undergoes dispersion. The following extends that analyses to a three-level system interacting with two photons. Analysis of the system is completed both numerically for all photonic field modes, and analytically for an adiabatic solution of a single field mode. The findings are not only interesting for understanding additional physical phenomena due to the increased complexity of a three-level, two-photon system, but are also necessary for advancing applications such as quantum communications, quantum computation, and quantum information.

## 1. Introduction

The electromagnetic field propagates through empty space at the speed of light, *c*. When the space is occupied by a medium, however, the speed may differ. A convenient way to account for the interaction between the radiation field and medium is through the index of refraction.

The index of refraction is used to quantify optical devices, and is traditionally based on macroscopic interactions. When the medium and radiation field consists of only a few particles, quantum phenomena must be taken into account. It has been shown [1] that the interaction of a single photon with a single two-level atom still exhibits the idea of an index of refraction.

The understanding of photonic interactions with an atom has implications in quantum information theory and quantum computations. It is well known that, due to the multiple degrees of freedom a photon has (spatial, temporal, and polarization), they can be used for superdense coding [2]. That is, a single photon can be used to encode multiple quantum bits (qubits) of information. Demonstration of two-qubit quantum operations on a single photon have been achieved [3]. There has also been experimental success that showed the manipulation of a three-qubit single photon state via linear deterministic quantum gates that exploited the photon’s polarization and two-dimensional spatial-parity-symmetry of the photon’s transverse field [2]. Quantum communications over long distances have also been demonstrated via satellites [4].

These revelations conclude that the propagation of photonic states are of great importance, such as in studying quantum state stability in which the state may be carrying quantum information for communications. Even more so, it is important to understand methods of being able to control a photon beam where the quantum nature cannot be ignored, such as when the number of photons is small.

In the following, it is shown that the phase shift created by a single atom on a two-photon field can be detected using a Mach–Zehnder interferometer. This is performed by first introducing the basic operators from cavity quantum electrodynamics, then developing the Hamiltonian of the interaction between the photon field and atom. The solution of this Hamiltonian is completed both numerically for multiple modes of the photon field, as well as analytically for a single mode under the adiabatic approximation. The basics of quantum optical instruments are then discussed to develop the Mach–Zehnder interferometer to which the solutions of the atom–field interaction are then transported through. Furthermore, introduction of a variable χ-phase changer within the interferometer displays the possibility of controlling the entangled states after the field–atom interaction. The results demonstrate that, as for a single photon with a two-level atom, a two-photon field with a three-level atom still contains idea of an index of refraction which can be implemented across many quantum fields, including quantum information, quantum computing, quantum communications, and even improving imaging in microscopy.

## 2. Atomic Interactions with Quantum Fields

The interaction of an electromagnetic field and atom are fundamentally quantum in nature. As such, the electronic state of the atom as well as the photonic state of the electromagnetic field are governed by *quantum operators*.

This section develops the necessary quantum operators needed to analyze a photonic field interacting with a general *n*-level atom. It should be noted that the development here is sequential, and serves as the beginning of the setup for the physical system which is to be incorporated with the Hamiltonian. These include parameters such as the field polarization and dynamic forms of the electromagnetic field.

### 2.1. The Atomic Transition Operator

Consider an *n*-level atom with $N_{e}$ bounded electrons. A given electron $e^{-}$ in this system starts in an initial state |i〉 and through some process ends in the final state |j〉, which is restricted by the Pauli exclusion principle via the Fermi–Dirac commutation relations:(1)$$e^{\frac{-iE_{i}t}{\hbar }}|i\rangle \overset{\begin{array}{c} \mathrm{Electronic}\\ \mathrm{Transition} \end{array}}{\to }e^{\frac{-iE_{j}t}{\hbar }}|j\rangle$$
where the phase associated with the state’s energy is displayed explicitly. Here, $E_{i}$ and $E_{j}$ are the electron’s initial and final energy, respectively. The operator associated with this state change is the atomic-transition operator ${\hat{\sigma }}_{ij}$, and is defined as(2)$${\hat{\sigma }}_{ij}=e^{-i\omega_{ij}t}|j\rangle \langle i|$$
such that $\omega_{ij}=\left(E_{j}-E_{i}\right)/\hbar$ is the driving frequency of the two atomic energy levels, which can be seen in Figure 1.

An important case of the atomic-transition operator is when the initial and final states are equal. In this case, ${\hat{\sigma }}_{ii}=|i\rangle \langle i|$ and leaves the state unchanged, even in phase. This is useful when the energy of a given electron is to be calculated.

### 2.2. The Electromagnetic Field Operator

The electromagnetic field operator will be developed under the cavity approximation [5]. In this approximation, a photon γ is restricted to confide in a perfectly conducting resonant cavity. In such a cavity, the field is required to obey the boundary conditions set forth by the Maxwell equations. The boundary conditions restrict the available wave vectors for the force mediating photon to exhibit, turning an integration over all wave vector space into a discrete sum over wave number space dependent on the cavity geometry.

Each wave vector the photon can exhibit is related to a *mode* of the field. The development of the electromagnetic field operator will be first performed for a single mode of a single photon in a cavity. Next, the electromagnetic field operator will be generalized for a single photon that has access to all modes of the given resonant cavity. Finally, the general electromagnetic field operator for a cavity will be given which includes multiple photons, each of which can access all available modes of the given cavity.

#### 2.2.1. Single-Mode Fields

A single mode-state of the classical electromagnetic field at the position ${\vec{x}}_{0}$ and time *t* in a rectangular cavity of volume *V* can be given as(3)$$\begin{array}{cc} {\vec{E}}_{\vec{\gamma }}\left({\vec{x}}_{0},t\right)= & \sum_{s}{\hat{\epsilon }}_{\vec{\gamma }}^{\left(s\right)}E_{\vec{\gamma }}\alpha_{\vec{\gamma }}^{\left(s\right)}e^{-i\nu_{\gamma }t+i\vec{\gamma }\cdot {\vec{x}}_{0}}\\ \; & +\sum_{s}{\hat{\epsilon }}_{\vec{\gamma }}^{*(s)}E_{\vec{\gamma }}\alpha_{\vec{\gamma }}^{*(s)}e^{i\nu_{\gamma }t-i\vec{\gamma }\cdot {\vec{x}}_{0}} \end{array}$$
where $\vec{\gamma }$ is the wave vector, ${\hat{\epsilon }}_{\vec{\gamma }}$ and ${\hat{\epsilon }}_{\vec{\gamma }}^{*}$ are the unit polarization vectors, $\alpha_{\vec{\gamma }}$ and $\alpha_{\vec{\gamma }}^{*}$ are amplitudes and dimensionless, $\nu_{\gamma }$ is the field frequency, *s* is the polarization state, and finally $E_{\vec{\gamma }}=\sqrt{\hbar \nu_{\gamma }/2\varepsilon_{0}V}$ give the units of an electric field. The associated magnetic fields can be obtained from $\vec{H}\left({\vec{x}}_{0},t\right)=\left[\vec{\gamma }\times \vec{E}\left({\vec{x}}_{0},t\right)\right]/\mu_{0}$, and will not be examined henceforth. The wave vector is given as $\vec{\gamma }=\langle \gamma_{x},\gamma_{y},\gamma_{z}\rangle$, which has restricted values from the boundary conditions as $\gamma_{x}=2\pi n_{x}/L_{x}$, $\gamma_{z}=2\pi n_{z}/L_{y}$, and $\gamma_{z}=2\pi n_{z}/L_{z}$ where $n_{x},n_{y},n_{z}\in Z$. A set $\left(n_{x},n_{y},n_{z}\right)$ defines a mode in the cavity approximation.

For simplification, the polarization basis vectors are chosen to be real, so that ${\hat{\epsilon }}_{\vec{\gamma }}={\hat{\epsilon }}_{\vec{\gamma }}^{*}$. Furthermore, the polarization state will be chosen to be linear. Thus, the electric field is now expressed as(4)$${\vec{E}}_{\vec{\gamma }}\left({\vec{x}}_{0},t\right)={\hat{\epsilon }}_{\vec{\gamma }}E_{\vec{\gamma }}\left(\alpha_{\vec{\gamma }}e^{-i\nu_{\gamma }t+i\vec{\gamma }\cdot {\vec{x}}_{0}}+\alpha_{\vec{\gamma }}e^{i\nu_{\gamma }t-i\vec{\gamma }\cdot {\vec{x}}_{0}}\right)$$
The classical electromagnetic field mode can be quantized simply by promotion of $\alpha_{\vec{\gamma }}$ and $\alpha_{\vec{\gamma }}^{*}$ to the harmonic creation and annihilation operators ${\hat{a}}_{\vec{\gamma }}^{+}$ and ${\hat{a}}_{\vec{\gamma }}$, respectively. As usual, the harmonic creation and annihilation operators obey the standard commutation relation $\left[{\hat{a}}_{\vec{\gamma }},{\hat{a}}_{\vec{\gamma }}^{+}\right]=1$. This results in the single mode electromagnetic field operator as(5)$${\hat{\vec{E}}}_{\vec{\gamma }}\left({\vec{x}}_{0},t\right)={\hat{\epsilon }}_{\vec{\gamma }}E_{\vec{\gamma }}\left({\hat{a}}_{\vec{\gamma }}e^{-i\nu_{\gamma }t+i\vec{\gamma }\cdot {\vec{x}}_{0}}+{\hat{a}}_{\vec{\gamma }}^{+}e^{i\nu_{\gamma }t-i\vec{\gamma }\cdot {\vec{x}}_{0}}\right)$$
Verification that this operator promotion is accurate can be achieved by inserting the electric and magnetic field mode equations into the classical electromagnetic field Hamiltonian and showing it is dynamically identical to a harmonic oscillator. The introduction of the creation and annihilation operators shows the field can create and destroy particles associated with it, namely photons.

It is typical, and convenient in some cases, to separate the positive and negative frequency parts of the electromagnetic field as(6)$${\hat{\vec{E}}}_{\vec{\gamma }}\left({\vec{x}}_{0},t\right)={\hat{\vec{E}}}_{\vec{\gamma }}^{\left(+\right)}\left({\vec{x}}_{0},t\right)+{\hat{\vec{E}}}_{\vec{\gamma }}^{\left(-\right)}\left({\vec{x}}_{0},t\right)$$
where ${\hat{\vec{E}}}_{\vec{\gamma }}^{\left(+\right)}\left({\vec{x}}_{0},t\right)$ is the positive frequency portion containing annihilation operators and ${\hat{\vec{E}}}_{\vec{\gamma }}^{\left(-\right)}\left({\vec{x}}_{0},t\right)$ is the negative frequency portion containing only creation operators. The forms of the frequency components can be seen explicitly as(7)$$\begin{array}{cc} {\hat{\vec{E}}}_{\vec{\gamma }}^{\left(+\right)}\left({\vec{x}}_{0},t\right) & ={\hat{\epsilon }}_{\vec{\gamma }}E_{\vec{\gamma }}{\hat{a}}_{\vec{\gamma }}e^{-i\nu_{\gamma }t+i\vec{\gamma }\cdot {\vec{x}}_{0}}\\ {\hat{\vec{E}}}_{\vec{\gamma }}^{\left(-\right)}\left({\vec{x}}_{0},t\right) & ={\hat{\epsilon }}_{\vec{\gamma }}E_{\vec{\gamma }}{\hat{a}}_{\vec{\gamma }}^{+}e^{i\nu_{\gamma }t-i\vec{\gamma }\cdot {\vec{x}}_{0}} \end{array}$$

This form of the electromagnetic field operator, given in Equation (Equation 6), shows the possibility of a single photon with a wave vector $\vec{\gamma }$ being created then destroyed throughout the cavity in a steady-state manner. To include dynamical motion, the photonic excitation of the electromagnetic field is to be enveloped in a Gaussian wave packet [6], as shown by(8)$$\begin{array}{cc} {\hat{E}}_{\vec{\gamma }}^{\left(x\right)}\left({\vec{x}}_{0},t\right) & ⟶{\hat{E}}_{\vec{\gamma }}^{\left(x\right)}\left({\vec{x}}_{0},t\right)e^{\frac{-1}{2L^{2}}{\left({\vec{x}}_{0}-ct_{\gamma }\hat{\gamma }\right)}^{2}}\\ {\hat{E}}_{\vec{\gamma }}^{\left(y\right)}\left({\vec{x}}_{0},t\right) & ⟶{\hat{E}}_{\vec{\gamma }}^{\left(y\right)}\left({\vec{x}}_{0},t\right)e^{\frac{-1}{2L^{2}}{\left({\vec{x}}_{0}-ct_{\gamma }\hat{\gamma }\right)}^{2}}\\ {\hat{E}}_{\vec{\gamma }}^{\left(z\right)}\left({\vec{x}}_{0},t\right) & ⟶{\hat{E}}_{\vec{\gamma }}^{\left(z\right)}\left({\vec{x}}_{0},t\right)e^{\frac{-1}{2L^{2}}{\left({\vec{x}}_{0}-ct_{\gamma }\hat{\gamma }\right)}^{2}} \end{array}$$
Here, ${\vec{x}}_{0}$ is the location within the cavity where the electromagnetic field is excited at t=0, the “universal” time such that $t_{\gamma }=t+t_{e}^{\left(\gamma \right)}$ is a temporal offset from the universal time due to some elapsed time, $t_{e}^{\left(\gamma \right)}$ from creation. $\hat{\gamma }=\langle \gamma_{x}^{\prime },\gamma_{y}^{\prime },\gamma_{z}^{\prime }\rangle$ is a unit vector identifying the direction the excitation begins to propagate as defined by the wave vector. The superscript variables, (x),(y), and (z) refer to the field’s Cartesian components. Here, the Gaussian envelopes are chosen because they are widely used in optics combining the mathematical elegance with physical practicality, providing stable, smooth, and efficient solutions for pulse generation, propagation, and manipulation. Additionally, many laser systems naturally produce pulses with Gaussian envelopes, as they arise from the fundamental mode of many optical cavities (e.g., TEM_00_ mode). Gaussian pulses are also naturally emitted in processes involving spontaneous or stimulated emission, where the gain medium supports such distributions. Although optical pulses may not always be perfectly Gaussian, many real-world pulse shapes (e.g., sech^2^ or Lorentzian) can be approximated by Gaussian envelopes, especially when simplicity and broad applicability are preferred. Hereon, when referencing the electromagnetic field operator, the dynamic version containing the Gaussian envelope for propagation will be used.

#### 2.2.2. Multimodal Fields

Thus far, only a single photon in a single mode of the electromagnetic field residing in a conducting cavity has been examined. In actuality, there are other modes of the electromagnetic field the photon can excite and occupy. The inclusion of these modes are necessary for understanding one of the most fundamental atomic processes of the atom–electronic relaxation with spontaneous photonic emission.

Inclusion of all of the modes a photon can occupy is straightforward: simply sum over all modes that satisfy the boundary conditions for the given cavity. More conveniently, since the modes and wave vectors for a cavity are intimately related, it suffices to simply sum over all allowable wave vectors for the cavity. Thus, the electromagnetic field operator for multiple modes is simply(9)$${\hat{\vec{E}}}_{\vec{\gamma }}\left({\vec{x}}_{0},t\right)⟶\sum_{\vec{\gamma }}{\hat{\vec{E}}}_{\vec{\gamma }}\left({\vec{x}}_{0},t\right)$$
This gives the multimode electromagnetic field operator as(10)$${\hat{\vec{E}}}_{\gamma }\left({\vec{x}}_{0},t\right)=\sum_{\vec{\gamma }}{\hat{\vec{E}}}_{\vec{\gamma }}\left({\vec{x}}_{0},t\right)$$
A slight change in notation has also been used. On the left hand side, the subscript γ is now used as an identifier for the γ-photon in the field, which can be in any of the $\vec{\gamma }$ modes of the electromagnetic field.

#### 2.2.3. Multiple Multimodal Fields

Seeing how a single photon may occupy any allowable mode of a resonant cavity, the next step is to include multiple photonic excitations, each able to access any mode of the electromagnetic field within the cavity. Inclusion of all excitations is once again as simple as summing over all photons within the cavity,(11)$$\begin{array}{cc} \hat{\vec{E}}\left({\vec{x}}_{0},t\right) & ={\hat{\vec{E}}}_{\gamma }\left({\vec{x}}_{0},t\right)+{\hat{\vec{E}}}_{\lambda }\left({\vec{x}}_{0},t\right)+\ldots \\ \; & =\sum_{\gamma }{\hat{\vec{E}}}_{\gamma }\left({\vec{x}}_{0},t\right) \end{array}$$
where the notation from Equation (Equation 10) has been used for photon labeling with each photon having its own set of available modes.

## 3. Hamiltonian of the System

The following details the solution of a three-level atom interacting with two photons. The dynamics of the electromagnetic field will be captured by treating the incoming photons via Gaussian wave-packets. It is noted that the atom interacting with the electromagnetic field is in a ladder scheme where $E_{a}<E_{b}<E_{c}$. Furthermore, the atomic states |a〉 and |b〉 interact via a photon *k*, and atomic states |b〉 and |c〉 interact via a photon *q*. The electron of the atom will initially be in the ground state, |a〉.

The Hamiltonian of such a system can be given as [5](12)$$\hat{H}={\hat{H}}_{atom}+{\hat{H}}_{field}-{\hat{V}}_{int}$$
In this, ${\hat{H}}_{atom}$ is the portion of the Hamiltonian associated with the electronic energy levels of the atom, ${\hat{H}}_{field}$ is related to the free energy in the electromagnetic field, and ${\hat{V}}_{int}$ is due to the interaction of the electromagnetic field and atomic electrons.

### 3.1. Electronic Energy

The portion of the Hamiltonian associated with the atomic energy levels has the explicit form(13)$${\hat{H}}_{atom}=\sum_{i}E_{i}{\hat{\sigma }}_{ii}$$
where the summation is over the atomic levels, $E_{i}$ is the energy of the $i^{\mathrm{th}}$ atomic level, and ${\hat{\sigma }}_{ii}$ is the atomic transition operator. It should be noted that atomic levels are complete, in the sense that $\sum_{i}{\hat{\sigma }}_{ii}=\left[1\right]$, where [1] is an i×i unit matrix.

For the 3-level atom being considered, this results in(14)$${\hat{H}}_{atom}=E_{a}{\hat{\sigma }}_{aa}+E_{b}{\hat{\sigma }}_{bb}+E_{c}{\hat{\sigma }}_{cc}$$

### 3.2. Field Energy

The field energy of the Hamiltonian can be determined by recognizing the electromagnetic field is dynamically equivalent to the simple harmonic oscillator. Thus, the energy associated with each photonic excitation of the electromagnetic field can be calculated via the creation and annihilation operators in the same manner as the simple harmonic oscillator. In the case of two photons, *k* and *q*, the result is(15)$${\hat{H}}_{field}=\sum_{\vec{\gamma }}\hbar \nu_{\gamma }\left({\hat{a}}_{\vec{\gamma }}^{+}{\hat{a}}_{\vec{\gamma }}+\frac{1}{2}\right)$$

### 3.3. Interaction Energy

The interaction energy of the electromagnetic field and atom is determined via the dot product of the atomic dipole moment and electromagnetic field. In the case of the three-level atom interacting with two photons, the interaction potential can be seen as(16)$${\hat{V}}_{int}=\hat{\vec{p}}\cdot \left({\hat{\vec{E}}}_{k}+{\hat{\vec{E}}}_{q}\right)$$
in which $\hat{\vec{p}}$ is the dipole moment operator of the atom. The form of $\hat{\vec{p}}$ can be discerned from quantizing the classical dipole,(17)$$\vec{p}=\sum_{m}e^{-}{\vec{r}}_{m}$$
where $e^{-}$ is the elementary electric charge of the electron and ${\vec{r}}_{m}$ is the position of the $m^{\mathrm{th}}$ electron in the atom relative to the nucleus. Quantization occurs using completeness and the projection operator as(18)$$\begin{array}{cc} \hat{\vec{p}} & =\sum_{m}\sum_{i}\sum_{j}e^{-}|j\rangle \langle j|{\vec{r}}_{m}|i\rangle \langle i|\\ & =\sum_{m}\sum_{i}\sum_{j}{\vec{℘}}_{ijm}{\hat{\sigma }}_{ij} \end{array}$$
The summations over *i* and *j* are the energy levels of the atom, while ${\vec{℘}}_{ijm}=e^{-}\langle j|{\vec{r}}_{m}|i\rangle$ can be thought of the quantum mechanical dipole moment of the $m^{\mathrm{th}}$ electron. The dipole moments will be used under the dipole approximation: the field due to the electronic charge is assumed to be evenly distributed about the atom and constant in structure.

The system under investigation will be restricted to a single electron. With Equation (Equation 18), the interaction portion of the Hamiltonian is now(19)$$\begin{array}{cc} {\hat{V}}_{int}=( & {\vec{℘}}_{aa}{\hat{\sigma }}_{aa}+{\vec{℘}}_{ab}{\hat{\sigma }}_{ab}+{\vec{℘}}_{ac}{\hat{\sigma }}_{ac}+\\ \; & {\vec{℘}}_{ba}{\hat{\sigma }}_{ba}+{\vec{℘}}_{bb}{\hat{\sigma }}_{bb}+{\vec{℘}}_{bc}{\hat{\sigma }}_{bc}+\\ \; & {\vec{℘}}_{ca}{\hat{\sigma }}_{ca}+{\vec{℘}}_{cb}{\hat{\sigma }}_{cb}+{\vec{℘}}_{cc}{\hat{\sigma }}_{cc})\cdot \left({\hat{\vec{E}}}_{k}+{\hat{\vec{E}}}_{q}\right) \end{array}$$
Equation (Equation 19) has many simplifications that occur. First, there are no “self-dipoles”. This results in all diagonal terms ${\vec{℘}}_{ii}\sigma_{ii}$ being identically zero. It is, however, expected for there to be “dipole symmetry”, which gives the relation ${\vec{℘}}_{ij}={\vec{℘}}_{ji}$. The final simplification is that ${\vec{℘}}_{ac}{\hat{\sigma }}_{ac}={\vec{℘}}_{ca}{\hat{\sigma }}_{ca}=0$. This is a result of the atomic transition selection rule, namely that Δl=±1, where *l* is the angular moment of the state. As an illustration, suppose |a〉 is in symmetrical *s* state (l=0). This means that |b〉 must be in the *p* state (l=1), which is antisymmetric. As a result, |c〉 is either in the *s* state (l=0) or *d* state (l=2), which are again symmetrical. This would mean that ${\vec{℘}}_{ac}∼{\vec{℘}}_{ss}={\vec{℘}}_{sd}=0$.

The prior discussion results in a reduction of Equation (Equation 19) to the following form:(20)$${\hat{V}}_{int}=\left[{\vec{℘}}_{ab}\left(\hat{\sigma_{ab}}+{\hat{\sigma }}_{ba}\right)+{\vec{℘}}_{bc}\left(\hat{\sigma_{bc}}+{\hat{\sigma }}_{cb}\right)\right]\cdot \left({\hat{\vec{E}}}_{k}+{\hat{\vec{E}}}_{q}\right)$$

It will further be assumed that the *k*-photon arrives to the atom before the *q*-photon and that the *k*-photon is prepared in an energy distribution matching the $\omega_{ab}$ energy level, while the *q*-photon is prepared matching the $\omega_{bc}$ energy level. Then, a decent approximation is that any interactions between the *k*-photon and the $\omega_{bc}$ energy level is negligible and similarly for the *q*-photon and the $\omega_{ab}$ energy level. This results in the *k*-photon being required to interact with the |a〉⟶|b〉 transition and the *q*-photon being required to interact with the |b〉⟶|c〉 transition. This assumption reduces the interaction Hamiltonian to(21)$$\begin{array}{cc} {\hat{V}}_{int} & ={\hat{E}}_{k}\left({\hat{\sigma }}_{ab}+{\hat{\sigma }}_{ba}\right)+{\hat{E}}_{q}\left({\hat{\sigma }}_{bc}+{\hat{\sigma }}_{cb}\right)\\ \; & =\left[{\hat{E}}_{k}^{\left(+\right)}+{\hat{E}}_{k}^{\left(-\right)}\right]\left({\hat{\sigma }}_{ab}+{\hat{\sigma }}_{ba}\right)\\ & +\left[{\hat{E}}_{q}^{\left(+\right)}+{\hat{E}}_{q}^{\left(-\right)}\right]\left({\hat{\sigma }}_{bc}+{\hat{\sigma }}_{cb}\right) \end{array}$$
where the electromagnetic field operator clearly commutes with the atomic transition operator. Also, the coupling constants(22)$$\begin{array}{cc} g_{\vec{k}} & =\left({\vec{℘}}_{ab}\cdot {\hat{\epsilon }}_{\vec{k}}\right)E_{\vec{k}}\\ g_{\vec{q}} & =\left({\vec{℘}}_{bc}\cdot {\hat{\epsilon }}_{\vec{q}}\right)E_{\vec{q}} \end{array}$$
have been introduced and absorbed into the appropriate electromagnetic field operator. It can be seen in Equation (Equation 21) that there are cross-terms that correspond to the creation of a photon *and* an increase in atomic energy level, such as ${\hat{E}}_{k}^{\left(-\right)}{\hat{\sigma }}_{ab}$. Likewise, there are cross-terms terms that correspond to the annihilation of a photon with a drop in atomic energy level, like ${\hat{E}}_{q}^{\left(+\right)}{\hat{\sigma }}_{cb}$. Such terms are non-energy-conserving, and are to be neglected under what is known as the *rotating wave approximation*. This gives the final form of the interaction potential as(23)$${\hat{V}}_{int}={\hat{E}}_{\vec{k}}^{\left(+\right)}{\hat{\sigma }}_{ab}+{\hat{E}}_{\vec{k}}^{\left(-\right)}{\hat{\sigma }}_{ba}+{\hat{E}}_{\vec{q}}^{\left(+\right)}{\hat{\sigma }}_{bc}+{\hat{E}}_{\vec{q}}^{\left(-\right)}{\hat{\sigma }}_{cb}$$

## 4. Equations of Motion

The equations of motion due to this Hamiltonian will be determined in the Schrodinger picture using probability amplitudes. Furthermore, interest is isolated to the interaction of the electromagnetic field and the atom. Thus, only the interaction potential will be examined and solved accordingly. The system will be assumed to be in a one dimensional cavity along the *z*-direction.

### 4.1. Probability Amplitudes

Beginning with the Schrodinger equation,(24)$$i\hbar \frac{\partial }{\partial t}|\Psi \rangle ={\hat{V}}_{int}|\Psi \rangle$$
the following assumed state with undetermined probability coefficients will be fed into Equation (Equation 24):(25)$$\begin{array}{cc} |\Psi \rangle = & C(t)|c\rangle ⊗|0_{k}\rangle ⊗|0_{q}\rangle +\\ \; & \sum_{q}B^{q}\left(t\right)|b\rangle ⊗|0_{k}\rangle ⊗|1_{q}\rangle +\\ \; & \sum_{q}\sum_{k}A_{k}^{q}\left(t\right)|a\rangle ⊗|1_{k}\rangle ⊗|1_{q}\rangle \end{array}$$
where summations occur due to the photons being able to be in any of the available modes.

The left hand side of Equation (Equation 24) applied to Equation (Equation 25) gives(26)$$\begin{array}{cc} i\hbar \frac{\partial }{\partial t}|\Psi \rangle = & i\hbar \dot{C}\left(t\right)|c,0_{k},0_{q}\rangle \\ + & i\hbar \sum_{q}{\dot{B}}^{q}\left(t\right)|b,0_{k},1_{q}\rangle \\ + & i\hbar \sum_{q}\sum_{k}{\dot{A}}_{k}^{q}\left(t\right)|a,1_{k},1_{q}\rangle \end{array}$$

Next is to apply Equation (Equation 23) to Equation (Equation 25) to obtain the right hand side of Equation (Equation 24). For the one dimensional setup along the *z*-axis, the electromagnetic field operators for a given photon γ have the explicit form(27)$$\begin{array}{cc} {\hat{E}}_{\gamma }^{\left(+\right)} & =\sum_{\gamma }g_{\gamma }{\hat{a}}_{\gamma }e^{-i\nu_{\gamma }t+i\gamma t}e^{\frac{1}{2L^{2}}{\left(z-ct\right)}^{2}}\\ {\hat{E}}_{\gamma }^{\left(-\right)} & =\sum_{\gamma }g_{\gamma }{\hat{a}}_{\gamma }^{+}e^{i\nu_{\gamma }t-i\gamma t}e^{\frac{1}{2L^{2}}{\left(z-ct\right)}^{2}} \end{array}$$
Applying Equation (Equation 23) to Equation (Equation 25) where the electromagnetic field operators have the form of Equation (Equation 27) will result in the right hand side of Equation (Equation 24) as(28)$$\begin{array}{cc} {\hat{V}}_{int}|\Psi \rangle = & \sum_{q}h_{q}^{*}\left(z,t_{d}\right)C\left(t\right)|b,0_{k},1_{q}\rangle \\ + & \sum_{q}h_{q}\left(z,t_{d}\right)B^{q}\left(t\right)|c,0_{q},0_{k}\rangle \\ + & \sum_{q}\sum_{k}h_{k}^{*}\left(z,t\right)B^{q}\left(t\right)|a,1_{q},1_{k}\rangle \\ + & \sum_{q}\sum_{k}h_{k}\left(z,t\right)A_{k}^{q}\left(t\right)|b,0_{k},1_{q}\rangle \end{array}$$
Comparison of the vector components of Equations (Equation 26) and (Equation 28) gives the following system of equations for their respective coefficients:(29)$$\left\{\begin{array}{cc} i\hbar \dot{C}\left(t\right) & =\sum_{q}h_{q}\left(z,t_{d}\right)B^{q}\left(t\right)\\ i\hbar {\dot{B}}^{q}\left(t\right) & =h_{q}^{*}\left(z,t_{d}\right)C\left(t\right)+\sum_{k}h_{k}\left(z,t\right)A_{k}^{q}\left(t\right)\\ i\hbar {\dot{A}}_{k}^{q}\left(t\right) & =h_{k}^{*}\left(z,t\right)B^{q}\left(t\right) \end{array}\right.$$
in which(30)$$\begin{array}{cc} h_{k}\left(z,t\right) & =g_{k}e^{ikz-i\Delta_{k}t}e^{\frac{-1}{2L^{2}}{\left(z-ct\right)}^{2}}\\ h_{q}\left(z,t_{d}\right) & =g_{q}e^{iqz-i\Delta_{q}t}e^{\frac{-1}{2L^{2}}{\left(z-ct_{d}\right)}^{2}} \end{array}$$
where $\Delta_{k}=\nu_{k}-\omega_{ab}$ and $\Delta_{q}=\nu_{q}-\omega_{bc}$ are the detuning factors, and $t_{d}=t+t_{e}$ is the delayed time that the *q*-photon arrives at the atom from a temporal spacing of some elapsed time, $t_{e}$. As usual, the (*) operator denotes complex conjugation.

### 4.2. Coupling and the Two-Level Atom

From Equation (Equation 29), it can be seen that |b〉 is coupled with both states |a〉 and |c〉, while the states |a〉 and |c〉 have no direct coupling, a result of the demonstrated angular momentum selection rule.

It is, in fact, possible to obtain the system of equations modeling a two-level system with a single electron interacting with a single photon from Equation (Equation 29). This is performed by decoupling the |b〉 and |c〉 states. Decoupling results in(31)$$\left\{\begin{array}{cc} i\hbar \dot{B}\left(t\right) & =\sum_{k}h_{k}\left(z,t\right)A_{k}\left(t\right)\\ i\hbar {\dot{A}}_{k}\left(t\right) & =h_{k}^{*}\left(z,t\right)B\left(t\right) \end{array}\right.$$
The solution to the two-level system [1] will be convenient for comparing similarities as well as new phenomena obtained from the higher complexity of the three-level system.

## 5. Adiabatic Solutions

To find an analytical solution, we use the mode function approach [6] that involves selecting an appropriate form for the mode. We consider a Gaussian mode function, which effectively consists of a superposition of many plane wave modes.

For a single mode, it is possible to obtain an analytic expression for B(t) under the adiabatic approximation, which is sufficient to determine dispersion results. As a result of obtaining B(t), similar expressions can be obtained for A(t) and C(t) by differentiation of the obtained equation, B(t). First, for a single mode, Equation (Equation 29) takes the form(32)$$\left\{\begin{array}{cc} i\hbar \dot{C}\left(t\right)= & h_{q}\left(z,t_{d}\right)B\left(t\right)\\ i\hbar \dot{B}\left(t\right)= & h_{k}\left(z,t\right)A\left(t\right)+h_{q}^{*}\left(z,t_{d}\right)C\left(t\right)\\ i\hbar \dot{A}\left(t\right)= & h_{k}^{*}\left(z,t\right)B\left(t\right) \end{array}\right.$$
From here, direct integration of $\dot{A}\left(t\right)$ and $\dot{C}\left(t\right)$ yields(33)$$\begin{array}{cc} C(t)= & \frac{-i}{\hbar }g_{q}e^{iqz}\int e^{-i\Delta_{q}t}f\left(z,t_{d}\right)B\left(t\right)dt\\ A(t)= & \frac{-i}{\hbar }g_{k}e^{-ikz}\int e^{i\Delta_{k}t}f\left(z,t\right)B\left(t\right)dt \end{array}$$
where the functions f(z,t) and $f(z,t_{d})$ are the Gaussian envelopes of $h_{k}\left(z,t\right)$ and $h_{q}\left(z,t_{d}\right)$, respectively. Equation (Equation 33) can now be substituted into the $\dot{B}\left(t\right)$ equation from Equation (Equation 32) to give(34)$$\begin{array}{cc} \dot{B}\left(t\right)= & \frac{-1}{\hbar^{2}}g_{k}^{2}e^{-i\Delta_{k}t}f\left(z,t\right)\int e^{i\Delta_{k}t}f\left(z,t\right)B\left(t\right)dt\\ \; & -\frac{1}{\hbar^{2}}g_{q}^{2}e^{i\Delta_{q}t}f\left(z,t_{d}\right)\int e^{-i\Delta_{q}t}f\left(z,t_{d}\right)B\left(t\right)dt \end{array}$$
Now, B(t) is expected to be well-behaved and, as such, may be integrated by parts to cast Equation (Equation 34) as(35)$$\begin{array}{cc} \dot{B}\left(t\right)= & \frac{-ig_{k}^{2}}{\hbar^{2}\Delta_{k}}f\left(z,t\right)\sum_{n=0}^{\infty }{\left(\frac{i}{\Delta_{k}}\right)}^{n}\frac{\partial^{n}}{\partial t^{n}}\left[f\left(z,t\right)B\left(t\right)\right]\\ \; & -\frac{ig_{q}^{2}}{\hbar^{2}\Delta_{q}}f\left(z,t_{d}\right)\sum_{m=0}^{\infty }{\left(\frac{-i}{\Delta_{q}}\right)}^{m}\frac{\partial^{m}}{\partial t^{m}}\left[f\left(z,t_{d}\right)B\left(t\right)\right] \end{array}$$
It is at this point the adiabatic approximation is invoked. This is performed by taking the first two terms in each of the summation expansions in Equation (Equation 35). Thus,(36)$$\begin{array}{cc} \dot{B}\left(t\right) & \approx \frac{ig_{k}^{2}}{\hbar^{2}\Delta_{k}}f\left(z,t\right)\left\{f\left(z,t\right)B\left(t\right)+\frac{i}{\Delta_{k}}\frac{\partial }{\partial t}\left[f\left(z,t\right)B\left(t\right)\right]\right\}\\ - & \frac{ig_{q}^{2}}{\hbar^{2}\Delta_{q}}f\left(z,t_{d}\right)\left\{f\left(z,t_{d}\right)B\left(t\right)-\frac{i}{\Delta_{q}}\frac{\partial }{\partial t}\left[f\left(z,t_{d}\right)B\left(t\right)\right]\right\} \end{array}$$
Equation (Equation 36) can be arranged into the form(37)$$\dot{B}\left(t\right)=\frac{\frac{-ig_{q}^{2}}{\Delta_{q}}f_{d}^{2}+\frac{ig_{k}^{2}}{\Delta_{k}}f^{2}-\frac{g_{k}^{2}}{\Delta_{k}^{2}}\dot{f}f-\frac{g_{q}^{2}}{\Delta_{q}^{2}}{\dot{f}}_{d}f_{d}}{\hbar^{2}+\frac{g_{k}^{2}}{\Delta_{k}^{2}}f^{2}+\frac{g_{q}^{2}}{\Delta_{q}^{2}}f_{d}^{2}}B\left(t\right)$$
where f=f(z,t) and $f_{d}=f\left(z,t_{d}\right)$. A solution to Equation (Equation 37) is given as(38)$$\begin{array}{c} B\left(t\right)=\frac{\hbar B_{0}}{\sqrt{\hbar^{2}+\frac{g_{k}^{2}}{\Delta_{k}^{2}}f^{2}\left(z,t\right)+\frac{g_{q}^{2}}{\Delta_{q}^{2}}f^{2}\left(z,t_{d}\right)}}\exp\left\{\frac{i\int dt\left[\frac{g_{k}^{2}}{\Delta_{k}}f^{2}\left(z,t\right)-\frac{g_{q}^{2}}{\Delta_{q}}f^{2}\left(z,t_{d}\right)\right]}{\left[\hbar^{2}+\frac{g_{k}^{2}}{\Delta_{k}^{2}}f^{2}\left(z,t\right)+\frac{g_{q}^{2}}{\Delta_{q}^{2}}f^{2}\left(z,t_{d}\right)\right]}\right\} \end{array}$$

$B_{0}$ in Equation (Equation 38) is the integration constant. With Equation (Equation 38) at hand, the field state A(t) may be calculated. Under the adiabatic approximation, the field state is(39)$$\begin{array}{cc} A(t)= & -\frac{g_{k}}{\hbar \Delta_{k}}e^{ikz-i\Delta_{k}t}\left\{fB\left(t\right)+\frac{i}{\Delta_{k}}\left[\dot{f}B\left(t\right)+\dot{B}\left(t\right)f\right]\right\}\\ \; & +A_{0} \end{array}$$
where *f* is the Gaussian envelope describing the *k*-photon stated previously, and $A_{0}$ is a constant of the integration.

It is possible to calculate the measured field at the detector. It is known [5] that the measured photon field at a detector is proportional to the first order correlation function $G^{\left(1\right)}$, where(40)$$G^{\left(1\right)}={\left|\langle 0_{k},0_{q}\left|{\hat{E}}^{\left(+\right)}\right|\Psi_{\gamma }\rangle \right|}^{2}$$
such that $|\Psi_{\gamma }\rangle$ represents the field portions of |Ψ〉, in particular(41)$$|\Psi_{\gamma }\rangle =C\left(t\right)|0_{k},0_{q}\rangle +B^{q}\left(t\right)|0_{k},1_{q}\rangle +A_{k}^{q}\left(t\right)|1_{k},1_{q}\rangle$$
In the realm of quantum applications, there is usually higher interest in the details of the phase of the measured quantum state. Letting $E=\langle 0_{k},0_{q}\left|\left({\hat{E}}_{k}^{\left(+\right)}+{\hat{E}}_{q}^{\left(+\right)}\right)\right|\Psi_{\gamma }\rangle$, where ${\hat{E}}^{\left(+\right)}$ has been expanded for the two photons, it can be shown that(42)$$E=g_{q}e^{iqz-i\nu_{q}t}f\left(z,t_{d}\right)B\left(t\right)$$
Determining the phase of E allows for the calculation of the phase and group velocities, letting(43)$$\delta_{k}^{q}\left(t\right)=\int dt\left[\frac{g_{k}^{2}}{\Delta_{k}}f^{2}\left(z,t\right)-\frac{g_{q}^{2}}{\Delta_{q}}f^{2}\left(z,t_{d}\right)\right]/\left[\hbar^{2}+\frac{g_{k}^{2}}{\Delta_{k}^{2}}f^{2}\left(z,t\right)+\frac{g_{q}^{2}}{\Delta_{q}^{2}}f^{2}\left(z,t_{d}\right)\right]$$
The phase of the field state $\chi_{k}^{q}\left(t\right)$ for a single mode under the adiabatic approximation can be given as(44)$$\chi_{k}^{q}=qz-\nu_{q}t+\delta_{k}^{q}\left(t\right)$$

Differentiation of $\chi_{k}^{q}$ with respect to time will yield the angular frequency which can then be used to calculate the phase and group velocities, as will be seen in Section 7.

## 6. The Quantum Beam Splitter and Mach–Zehnder Interferometer

Interferometry can be used to experimentally measure the dispersion of light. To observe the dispersion effects of individual photons, a Mach–Zehnder interferometer should be used. Such an interferometer is composed of two quantum beam splitters and two mirrors. A brief summary of the concepts and key equations for the quantum beam splitter and Mach–Zehnder interferometer are to be presented followed by the results of a three-level atom in one arm of the interferometer and a phase changer in the second arm.

### 6.1. Optical Instruments

The quantum beam splitter, Figure 2, is considered a four-port optical device with two input modes and two output modes [7]. The two input modes will be taken to be $|\alpha_{1}\rangle$ and $|\alpha_{2}\rangle$, while the two output modes will be $|\beta_{1}\rangle$ and $|\beta_{2}\rangle$.

The action of the beam splitter on the input states $|\alpha_{1}\rangle$ and $|\alpha_{2}\rangle$ can be represented by a general unitary matrix, [U]. The form of such a matrix is well known, and is given as(45)$$\left[U\right]=e^{i\eta }\left[\begin{array}{cc} e^{i\xi } & 0\\ 0 & e^{-i\xi } \end{array}\right]\left[\begin{array}{cc} \cos\theta & \sin\theta \\ -\sin\theta & \cos\theta \end{array}\right]\left[\begin{array}{cc} e^{i\phi } & 0\\ 0 & e^{-i\phi } \end{array}\right]$$
where η,ξ,θ,ϕ∈R. This form of [U] demonstrates that the four-port beam splitter is a three action process. First, the input modes undergo a phase change. Second, the amplitudes of the states are then rotated. Lastly, a final phase shift occurs.

The Schrodinger representation of [U] is obtained via the Jordan–Schwinger formulation of an angular momentum operator with two bosonic mode (creation/annihilation) operators,(46)$$\begin{array}{cc} {\hat{L}}_{0} & =\frac{1}{2}\left({\hat{\alpha }}_{1}^{+}{\hat{\alpha }}_{1}+{\hat{\alpha }}_{2}^{+}{\hat{\alpha }}_{2}\right)\\ {\hat{L}}_{1} & =\frac{1}{2}\left({\hat{\alpha }}_{1}^{+}{\hat{\alpha }}_{2}+{\hat{\alpha }}_{2}^{+}{\hat{\alpha }}_{1}\right)\\ {\hat{L}}_{2} & =\frac{-i}{2}\left({\hat{\alpha }}_{1}^{+}{\hat{\alpha }}_{2}-{\hat{\alpha }}_{2}^{+}{\hat{\alpha }}_{1}\right)\\ {\hat{L}}_{3} & =\frac{1}{2}\left({\hat{\alpha }}_{1}^{+}{\hat{\alpha }}_{1}-{\hat{\alpha }}_{2}^{+}{\hat{\alpha }}_{2}\right) \end{array}$$
such that the operators obey the commutation relations of the angular momentum operators. Physically, the $\hat{L}$-operators detail the spin properties of the light under investigation. It is possible to obtain the output states from these operators as(47)$$\left[\begin{array}{c} |\beta_{1}\rangle \\ |\beta_{2}\rangle \end{array}\right]=\hat{S}\left[\begin{array}{c} |\alpha_{1}\rangle \\ |\alpha_{2}\rangle \end{array}\right]{\hat{S}}^{+}$$
such that(48)$$\hat{S}=e^{-i\phi {\hat{L}}_{3}}e^{-i\theta {\hat{L}}_{2}}e^{-i\xi {\hat{L}}_{3}}e^{-i\eta {\hat{L}}_{0}}$$
By explicit calculation, the creation and annihilation operators of the input states can be expressed in terms of the creation and annihilation operators of the output states. Doing so gives(49)$$\begin{array}{cc} {\hat{\alpha }}_{1}\left({\hat{\beta }}_{1},{\hat{\beta }}_{2}\right)= & {\hat{\beta }}_{1}e^{-i(\phi +\xi +\eta )}\cos\theta -{\hat{\beta }}_{2}e^{i(-\phi +\xi -\eta )}\sin\theta \\ {\hat{\alpha }}_{2}\left({\hat{\beta }}_{1},{\hat{\beta }}_{2}\right)= & {\hat{\beta }}_{1}e^{i(\phi -\xi -\eta )}\sin\theta +{\hat{\beta }}_{2}e^{i(\phi +\xi -\eta )}\cos\theta \\ {\hat{\alpha }}_{1}^{+}\left({\hat{\beta }}_{1}^{+},{\hat{\beta }}_{2}^{+}\right)= & {\hat{\beta }}_{1}^{+}e^{i(\phi +\xi +\eta )}\cos\theta -{\hat{\beta }}_{2}^{+}e^{-i(-\phi +\xi -\eta )}\sin\theta \\ {\hat{\alpha }}_{2}^{+}\left({\hat{\beta }}_{1}^{+},{\hat{\beta }}_{2}^{+}\right)= & {\hat{\beta }}_{1}^{+}e^{-i(\phi -\xi -\eta )}\sin\theta +{\hat{\beta }}_{2}^{+}e^{-i(\phi +\xi -\eta )}\cos\theta \end{array}$$

The use of two quantum beams splitters and two mirrors can create the Mach–Zehnder interferometer, the setup for which is shown in Figure 3, and includes the three-level atom in the upper arm and a variable phase changer in the lower arm. Figure 3 serves as a theoretical setup, and a brief discussion at the end of this section concerns experimental implementation.

An empty Mach–Zehnder interferometer, one without an atom or phase changer, is balanced. Thus, the probability the photons choose either arm is equal. Of particular interest in the interferometer will be the interaction of two photons with a three-level atom; thus, examination of two photons in the empty interferometer is to be performed first.

There are four possible input states; however, by symmetry and index rearrangement only two need to be examined, namely(50)$$\begin{array}{cc} |I_{1}\rangle & =|\alpha_{1}\rangle ⊗|\alpha_{2}\rangle \\ \; & =|\left\{1_{k},1_{q}\right\},\left\{0_{k},0_{q}\right\}\rangle \\ \; & ={\hat{\alpha }}_{1_{k}}^{+}{\hat{\alpha }}_{1_{q}}^{+}|\left\{0_{k},0_{q}\right\},\left\{0_{k},0_{q}\right\}\rangle \end{array}$$
which represents both input photons, in the single modes *k* and *q*, entering through the same port of the first beam splitter. In this case, port $|\alpha_{1}\rangle$. The second possible input state is(51)$$\begin{array}{cc} |I_{2}\rangle & =|\alpha_{1}\rangle ⊗|\alpha_{2}\rangle \\ \; & =|\left\{1_{k},0_{q}\right\},\left\{0_{k},1_{q}\right\}\rangle \\ \; & ={\hat{\alpha }}_{1_{k}}^{+}{\hat{\alpha }}_{2_{q}}^{+}|\left\{0_{k},0_{q}\right\},\left\{0_{k},0_{q}\right\}\rangle \end{array}$$
which represents one photon entering one port while the other photon enters through the second port. In this case, the *k* photon enters through port $|\alpha_{1}\rangle$ whereas the *q* photon enters via the $|\alpha_{2}\rangle$ port. Application of Equation (Equation 49) to Equations (Equation 50) and (Equation 51) gives the propagation states, $|P_{1}\rangle$ and $|P_{2}\rangle$, in the upper and lower arms of the interferometer, respectively, as(52)$$\begin{array}{cc} |P_{1}\rangle = & |\beta_{1}\rangle ⊗|\beta_{2}\rangle \\ = & {\hat{\alpha }}_{1_{k}}^{+}\left({\hat{\beta }}_{1_{k}}^{+},{\hat{\beta }}_{2_{k}}^{+}\right){\hat{\alpha }}_{1_{q}}^{+}\left({\hat{\beta }}_{1_{q}}^{+},{\hat{\beta }}_{2_{q}}^{+}\right)|\left\{0_{k},0_{q}\right\},\left\{0_{k},0_{q}\right\}\rangle \\ = & e^{4i(\phi +\xi -3\eta )}\cos^{2}\theta |\left\{1_{k},1_{q}\right\},\left\{0_{k},0_{q}\right\}\rangle \\ \; & -e^{4i(\phi +\eta )}\cos\theta \sin\theta |\left\{1_{k},0_{q}\right\},\left\{0_{k},1_{q}\right\}\rangle \\ \; & +e^{-4i(-\phi +\xi -\eta )}\sin^{2}\theta |\left\{0_{k},0_{q}\right\},\left\{1_{k},1_{q}\right\}\rangle \\ \; & -e^{4i(\phi +\eta )}\cos\theta \sin\theta |\left\{0_{k},1_{q}\right\},\left\{1_{k},0_{q}\right\}\rangle \end{array}$$
while the second arrangement, having one photon in each input port, gives(53)$$\begin{array}{cc} |P_{2}\rangle = & |\beta_{1}\rangle ⊗|\beta_{2}\rangle \\ = & {\hat{\alpha }}_{1_{k}}^{+}\left({\hat{\beta }}_{1_{k}}^{+},{\hat{\beta }}_{2_{k}}^{+}\right){\hat{\alpha }}_{2_{q}}^{+}\left({\hat{\beta }}_{1_{q}}^{+},{\hat{\beta }}_{2_{q}}^{+}\right)|\left\{0_{k},0_{q}\right\},\left\{0_{k},0_{q}\right\}\rangle \\ = & e^{-6i\eta }\cos^{2}\theta |\left\{1_{k},0_{q}\right\},\left\{0_{k},1_{q}\right\}\rangle \\ \; & +e^{2i(\xi +\eta )}\cos\theta \sin\theta |\left\{1_{k},1_{q}\right\},\left\{0_{k},0_{q}\right\}\rangle \\ \; & -e^{2i\eta }\sin^{2}\theta |\left\{0_{k},1_{q}\right\},\left\{1_{k},0_{q}\right\}\rangle \\ \; & -e^{2i(-\xi +\eta )}\cos\theta \sin\theta |\left\{0_{k},0_{q}\right\},\left\{1_{k},1_{q}\right\}\rangle \end{array}$$
For studying the three-level atom, it is necessary for *both* photons to go through the same arm of the interferometer in order to excite up to the third atomic level, |c〉. This means that which photonic mode is in which arm is unimportant. What is important is that the photon *is* in a given arm. Then, $|2,0\rangle =|\left\{1_{k},1_{q}\right\},\left\{0_{k},0_{q}\right\}\rangle$, $|0,2\rangle =|\left\{0_{k},0_{q}\right\},\left\{1_{k},1_{q}\right\}\rangle$, and $|1,1\rangle =|\left\{1_{k},0_{q}\right\},\left\{0_{k},1_{q}\right\}\rangle =|\left\{0_{k},1_{q}\right\},\left\{1_{k},0_{q}\right\}\rangle$. It will also be assumed that the beam splitters are 50:50, which means cosθ=sinθ, resulting in θ=π/4. Then, the propagation states after the first beam splitter can be given as(54)$$\begin{array}{cc} |P_{1}\rangle = & \frac{1}{2}e^{4i(\phi +\xi -3\eta )}|2,0\rangle -\frac{1}{2}e^{-4i(-\phi +\xi -\eta )}|0,2\rangle \\ \; & -\frac{\sqrt{2}}{2}e^{4i(\phi +\eta )}|1,1\rangle \end{array}$$
and the second configuration results in(55)$$\begin{array}{cc} |P_{2}\rangle = & \frac{1}{2}e^{2i(\xi +\eta )}|2,0\rangle -\frac{1}{2}e^{2i(-\xi +\eta )}|0,2\rangle \\ \; & +\frac{1}{2}\left(e^{-6i\eta }-e^{2i\eta }\right)|1,1\rangle \end{array}$$
It can be seen that, for $|P_{1}\rangle$, there is always a 50% chance of the photons splitting into two separate arms. For $|P_{2}\rangle$, however, by choosing a *real* beam splitter, one that does not cause an overall rotation of the photon state (i.e., ϕ=ξ=η=0), the photons can be forced to emerge in pairs of the balanced/empty interferometer,(56)$$|P_{2}\rangle =\frac{1}{\sqrt{2}}|2,0\rangle -\frac{1}{\sqrt{2}}|0,2\rangle$$
where the states have been renormalized appropriately. With this in mind, only the input state $|I_{2}\rangle$ (now called |I〉) is to be used with real 50:50 beam splitters in the interferometer. Similarly to before, the output state $|O\rangle =|\zeta_{1}\rangle ⊗|\zeta_{1}\rangle$, after the second beam splitter can be determined. By applying [U] twice on the input state |I〉,(57)$$|O\rangle =\left[\begin{array}{c} U \end{array}\right]\left[\begin{array}{c} U \end{array}\right]|I\rangle$$
which, in terms of the input state, can be given as(58)$${\left[\begin{array}{c} U \end{array}\right]}^{-1}{\left[\begin{array}{c} U \end{array}\right]}^{-1}|O\rangle =|I\rangle$$
gives the creation operators of ${\hat{\alpha }}_{1}^{+}$ and ${\hat{\alpha }}_{2}^{+}$ in terms of ${\hat{\zeta }}_{1}^{+}$ and ${\hat{\zeta }}_{2}^{+}$ as(59)$$\begin{array}{cc} {\hat{\alpha }}_{1}^{+} & =-{\hat{\zeta }}_{2}^{+}\\ {\hat{\alpha }}_{2}^{+} & ={\hat{\zeta }}_{1}^{+} \end{array}$$
Thus, by inspection, if the photons go in two separate ports, after two real 50:50 beam splitters, the photons come out two separate ports. It can also be noted that the photons switched ports, and the photon from the first input port obtained a phase change of π,(60)$$\begin{array}{cc} |O\rangle & =-|1,1\rangle \\ \; & =-|\left\{0_{k},1_{q}\right\},\left\{1_{k},0_{q}\right\}\rangle \end{array}$$
where the modes are shown to display the interchanging of the photons as compared with Equation (Equation 51).

### 6.2. The Three-Level Atom and Interferometer

Introduction of the atom changes the probability by being a potential barrier for the photons, thus increasing their chance of going through the lower arm. The balance of the interferometer can be restored as well as varied by including a phase changer in the lower arm, as shown in Figure 3. The variation in this phase changer will, in turn, change the probability of which arm the photons are in, which upon detection details the dispersion. Inclusion of a χ-phase changer in the lower arm of the interferometer can be implemented with(61)$$\left[\begin{array}{c} \chi \end{array}\right]=\left[\begin{array}{cc} 1 & 0\\ 0 & e^{i\chi } \end{array}\right]$$
Then, the output state is obtained as(62)$$|O_{\chi }\rangle =\left[\begin{array}{c} U \end{array}\right]\left[\begin{array}{c} \chi \end{array}\right]\left[\begin{array}{c} U \end{array}\right]|I\rangle$$
Once again, this can be used to obtain(63)$$\begin{array}{cc} {\hat{\alpha }}_{1}^{+}=\frac{1}{2}\left[\left(1-e^{i\chi }\right){\hat{\zeta }}_{1}^{+}-\left(1+e^{i\chi }\right){\hat{\zeta }}_{2}^{+}\right] & \\ {\hat{\alpha }}_{2}^{+}=\frac{1}{2}\left[\left(1+e^{i\chi }\right){\hat{\zeta }}_{1}^{+}+\left(e^{i\chi }-1\right){\hat{\zeta }}_{2}^{+}\right] \end{array}$$
which, when applied on the input state |I〉 yields the output of a Mach–Zehnder interferometer with a phase changer in the lower arm as(64)$$\begin{array}{cc} |O_{\chi }\rangle = & \frac{1}{4}\left(1-e^{2i\chi }\right)|\left\{1_{k},1_{q}\right\},\left\{0_{k},0_{q}\right\}\rangle \\ \; & +\frac{1}{4}\left(1-e^{i\chi }\right)\left(e^{i\chi }-1\right)|\left\{1_{k},0_{q}\right\},\left\{0_{k},1_{q}\right\}\rangle \\ \; & -\frac{1}{4}{\left(1+e^{i\chi }\right)}^{2}|\left\{0_{k},1_{q}\right\},\left\{1_{k},0_{q}\right\}\rangle \\ \; & -\frac{1}{4}\left(e^{2i\chi }-1\right)|\left\{0_{k},0_{q}\right\},\left\{1_{k},1_{q}\right\}\rangle \end{array}$$
The photonic field modes have been shown explicitly to display the effect the χ-phase changer has on the interchanging of of the photons at the output. Interestingly enough, a χ-phase changer with χ=π results in the photons exiting from the same port modes as they entered.

Before placing the atom in the top arm of the interferometer, it is worth considering an interferometer with two phase changers, one in each arm. This is to contrast how a phase changer, which will change the phase of any photon by the same amount, contrasts with an atom, which is expected to change the photon field phase based on the mode state of the photons. In this case, $\left[\begin{array}{c} \chi \end{array}\right]⟶\left[\begin{array}{c} \chi_{0} \end{array}\right]$, where(65)$$\left[\begin{array}{c} \chi_{0} \end{array}\right]=\left[\begin{array}{cc} e^{i\chi_{0}} & 0\\ 0 & e^{i\chi } \end{array}\right]$$
which represents a $\chi_{0}$-phase changer in the top arm and a χ-phase changer in the bottom arm. As before,(66)$$|O_{\chi_{0}}\rangle =\left[\begin{array}{c} U \end{array}\right]\left[\begin{array}{c} \chi_{0} \end{array}\right]\left[\begin{array}{c} U \end{array}\right]|I\rangle$$
which, by the same method used prior, yields(67)$$\begin{array}{cc} {\hat{\alpha }}_{1}^{+} & =\frac{1}{2}\left[\left(e^{i\chi_{0}}-e^{i\chi }\right){\hat{\zeta }}_{1}^{+}-\left(e^{i\chi_{0}}+e^{i\chi }\right){\hat{\zeta }}_{2}^{+}\right]\\ {\hat{\alpha }}_{2}^{+} & =\frac{1}{2}\left[\left(e^{i\chi_{0}}+e^{i\chi }\right){\hat{\zeta }}_{1}^{+}+\left(e^{i\chi }-e^{i\chi_{0}}\right){\hat{\zeta }}_{2}^{+}\right] \end{array}$$
which gives the output state as(68)$$\begin{array}{cc} |O_{\chi_{0}}\rangle = & \frac{1}{4}\left(e^{2i\chi_{0}}-e^{2i\chi }\right)|\left\{1_{k},1_{q}\right\},\left\{0_{k},0_{q}\right\}\rangle \\ \; & +\frac{1}{4}\left(e^{i\chi_{0}}-e^{i\chi }\right)\left(e^{i\chi }-e^{i\chi_{0}}\right)|\left\{1_{k},0_{q}\right\},\left\{0_{k},1_{q}\right\}\rangle \\ \; & -\frac{1}{4}{\left(e^{i\chi_{0}}+e^{i\chi }\right)}^{2}|\left\{0_{k},1_{q}\right\},\left\{1_{k},0_{q}\right\}\rangle \\ \; & -\frac{1}{4}\left(e^{2i\chi }-e^{2i\chi_{0}}\right)|\left\{0_{k},0_{q}\right\},\left\{1_{k},1_{q}\right\}\rangle \end{array}$$
With the $\chi_{0}$-phase changer being replaced with an atom, it is necessary to solve Equation (Equation 29) under the interferometer setup shown in Figure 3. For an analytic demonstration, this will be performed under the adiabatic approximation for photons in a single mode, and then generalized to the multimode field. Numerical results will be presented for the multimode solution. The solution to Equation (Equation 32) is formally given as(69)$$\begin{array}{cc} |\Psi \rangle & =\hat{T}\exp\left[-\frac{i}{\hbar }\int_{t_{0}}^{t}{\hat{V}}_{int}\left(\tau \right)d\tau \right]|\Psi_{0}\rangle \\ \; & \approx e^{i\chi_{k}^{q}}|\Psi_{0}\rangle \end{array}$$
such that $\hat{T}$ is the time ordering operator to take into account that ${\hat{V}}_{int}\left(\tau_{1}\right)$ and ${\hat{V}}_{int}\left(\tau_{2}\right)$ may not commute. Then, Equation (Equation 39) has the solution(70)$$A\left(t\right)=A_{0}e^{i\chi_{k}^{q}}$$
Thus, if the photon field state started without any phase, it will have obtained the phase $\chi_{k}^{q}$ upon interacting with the atom. The form of $\chi_{k}^{q}$ is given in Equation (Equation 44), where the integral of $\delta_{k}^{q}\left(t\right)$ is to be taken over all time due to the fact that the final phase of the photon field, long after interaction with the atom, is what is to be measured/examined. From this, it is clear the atom acts similarly to the $\chi_{0}$-phase changer, with one stark distinction—the obtained phase change is dependent on the mode state of the photon field.

The output from the Mach–Zehnder interferometer with an atom in the upper arm and χ-phase changer in the lower arm is then given as(71)$$\begin{array}{cc} |O_{\chi_{k}^{q}}\rangle = & \frac{1}{4}\left(e^{2i\chi_{k}^{q}}-e^{2i\chi }\right)|\left\{1_{k},1_{q}\right\},\left\{0_{k},0_{q}\right\}\rangle \\ + & \frac{1}{4}\left(e^{i\chi_{k}^{q}}-e^{i\chi }\right)\left(e^{i\chi }-e^{i\chi_{k}^{q}}\right)|\left\{1_{k},0_{q}\right\},\left\{0_{k},1_{q}\right\}\rangle \\ - & \frac{1}{4}{\left(e^{i\chi_{k}^{q}}+e^{i\chi }\right)}^{2}|\left\{0_{k},1_{q}\right\},\left\{1_{k},0_{q}\right\}\rangle \\ - & \frac{1}{4}\left(e^{2i\chi }-e^{2i\chi_{k}^{q}}\right)|\left\{0_{k},0_{q}\right\},\left\{1_{k},1_{q}\right\}\rangle \end{array}$$
The generalization to the multimode nature of the photon field will be performed approximately. The approximation is that the mode probability distribution will remain unchanged due to the interaction of the atom, and will only receive a phase change. This amounts to treating $\left|A_{k}^{q}\left(t\right)\right|$ in Equation (Equation 25) as constant. Under this assumption, the multimode output from the Mach–Zehnder interferometer with an atom in the upper arm and a χ-phase changer in the lower arm is given by summing over all modes in Equation (Equation 71).(72)$$|O\rangle =\sum_{k}\sum_{q}A_{k}^{q}|O_{\chi_{k}^{q}}\rangle$$

The proposed experiment can be implemented using a Mach–Zehnder interferometer with a single-atom trap placed in one arm of the interferometer. Experimental demonstrations of single-atom trapping have been well established. For example, a single atom can be confined in a superoscillatory optical trap [8], or individual 87Rb atoms can be loaded from a magneto-optical trap into an optical dipole trap, operating with a far detuning of 61 nm from the atomic resonance [9]. In order to increase coupling with trapped atoms, the system of focusing mirrors can be used inside the Mach–Zehnder interferometer [10].

## 7. Numerical Results

Demonstration of the photonic dispersion via a single three-level atom can be seen from solving Equation (Equation 29). Results from a numerical evaluation of Equation (Equation 29) are presented below. The evaluation was completed using *SciPy’s* explicit Runge–Kutta fifth-order accurate formula of quartic interpolation polynomials, “*RK45*”. The absolute error and relative error of the solver were set to $10^{-8}$.

The simulation was completed in natural atomic units, such that Planck’s constant ℏ=1 and the speed of light c=1, and the permittivity of free space $\varepsilon_{0}=1$. As such, quantities of interest in the S.I. system are obtained by the following conversions: $\left[\mathrm{Time}\right]=\hbar /m_{e}c^{2}$, $\left[\mathrm{Length}\right]=\hbar /m_{e}c$, $\left[\mathrm{Mass}\right]=m_{e}$, [Velocity]=c, and $\left[\mathrm{Energy}\right]=m_{e}c^{2}$. The energy levels of the atom were set to match the scale of the valence electron of a rubidium atom, while the *k*-photon field had 275 modes available and the *q*-photon field had 175 modes available. The length of the cavity was chosen such that the $200^{\mathrm{th}}$ mode of the *k*-photon field was was in resonance with the $\omega_{ab}$ transition. The value for the length and mode are obtainable from the eigenfrequencies of the cavity k=2πn/L. With c=1, the wavenumber is equivalent to the frequency, which is set to $\omega_{ab}$ and n=200. This puts the resonance mode of the $\omega_{bc}$ transition near the $121^{\mathrm{st}}$ cavity mode.

The initial states of the photon fields were set such that the neighboring 40 modes above and below the resonance mode had equal probabilities. For the *k*-field, this corresponded to modes 160–240, and for the *q*-field, modes 80–160. The initial probability values were chosen to be real, and to have an initial phase of zero. The choice of implementing quantities similar to a rubidium atom comes from their ability to be trapped and contained in quantum electrodynamic cavities [10,11]. Furthermore, the details of the rubidium atom, such as its energy levels, transition frequencies, and coupling strengths, are known in detail [12].

Figure 4 demonstrates the phase change of a given mode of the photon field upon interaction with a three-level atom. The field starts with a near-linear phase change in time, upon which is changed during interaction, and then remains nearly linear in time after interaction. This demonstration shows the atom does indeed act as a phase changer. The linear aspects before and after interaction of the phase are also expected, since each mode in Equation (Equation 27) has a linear dependence in time without any interactions. Furthermore, with the phase change of multiple modes displayed, it can be seen quite readily how the photonic field phase change is dependent on the occupied mode. A demonstration that, unlike with a standard phase changer (which would change the phases of all modes of the photon field equally), the atom is mode dependent. The set of modes shown are $A_{1}^{1}\left(t\right)$, $A_{10}^{1}\left(t\right)$, $A_{20}^{1}\left(t\right)$, $A_{30}^{1}\left(t\right)$, $A_{40}^{1}\left(t\right)$, and $A_{50}^{1}\left(t\right)$.

As one can see from Equation (Equation 37), the amplitude B(t) acquires the phase(73)$$B\left(t\right)≃\exp\left(\frac{-ig_{q}^{2}}{\hbar^{2}\Delta_{q}}\int dtf_{d}^{2}+\frac{ig_{k}^{2}}{\hbar^{2}\Delta_{k}}\int dtf^{2}\right)B_{0},$$
that aligns with the results [1] and extends them to the two-photon regime. Similarly, for three-level atoms, the phase and group velocities can be defined as follows:(74)$$V_{ph}=\frac{\omega }{k}=c+c\left(\frac{-g_{q}^{2}}{\hbar^{2}\Delta_{q}\omega_{0}}f_{d}^{2}+\frac{g_{k}^{2}}{\hbar^{2}\Delta_{k}\omega_{0}}f^{2}\right),$$(75)$$V_{g}=\frac{\partial \omega }{\partial k}=c+c\left(\frac{-g_{q}^{2}}{\hbar^{2}\Delta_{q}^{2}}f_{d}^{2}+\frac{g_{k}^{2}}{\hbar^{2}\Delta_{k}^{2}}f^{2}\right).$$
The importance of this extension is that the three-level atom has different phase shift than the two-level atom, and it means that the dispersion of the three-level atom can provide separation the two-photon states from a single-photon state using the Mach–Zehnder interferometer.

Figure 5 demonstrates how, by tuning the χ-phase changer in the lower arm of the Mach–Zehnder interferometer, it is possible to control the output state. In particular, with each input of the interferometer having one of the photons, it is possible to tune the phase changer such that there is a large probability to obtain entangled output states. It can also be seen that the probability of |2,0〉=|0,2〉.

While the main focus in this paper is in understanding the phase change of the photon field due to atomic interactions, it is still interesting to see the relative state probabilities throughout the interaction. Figure 6 and Figure 7 show the probabilities of the electron to be in a given energy level and the existence of a photon in the field, respectively.

Figure 8 demonstrates the change in the field’s mode probability upon interaction with an atom. This phenomenon is not taken into account for the multimode output of the Mach–Zehnder interferometer. The initial condition is that $A_{k}^{1}\left(0\right)=0$ for all *k* of the *k*-photon. It can be seen clearly here that, at the time of interaction, there is now probability for the *q*-field to be in its first mode, demonstrated by *k*-photon probabilities appearing.

Figure 9 demonstrates the conservation of probability across the simulation of the two photons with a three-level atom. This is obtained by squaring and summing all coefficients from the solution of Equation (Equation 29). The probability remains at unity. Deviations from this are attributed to error tolerances of the simulation.

## 8. Conclusions

In closing, the completed analysis focused on the dynamic interaction of a two-photon field interacting with a single electron of a three-level atom. Aspects of the analysis studied included how to account for photonic propagation and interactions of a multiphoton field that has access to all available modes with an atom. The interaction of the field with the atom was shown to change the mode distribution of the photons, as well as induce a phase change upon the field states. Our approach even includes interactions with all vacuum modes, and the obtained results are valid for a mode function approach [6] when the photon has large detuning from the atomic resonance and the role of relaxation is negligible. From this, the phase and group velocities were obtained, and the results were contrasted to the two-level atom [1]. The importance of the phase properties were highlighted with respect to various other quantum fields, including quantum communications, quantum information, and quantum computing.

Insight into the dynamics of the two-photon field is critical when dealing with quantum information, where the photons themselves contain information in the form of qubits. By examining the field’s evolution across the interaction with a three-level atom, perception is gained into how the behavior and properties of the photon change.

A key aspect examined includes the analysis of the phase velocity of the photon field. The dispersion of light details the speed at which the phase of the field state propagates in space, and is thus responsible for the temporal behavior of the field state. As such, the phase velocity, and thus the dispersion, is imperative when analyzing protocols of quantum communication systems. A deeper understanding of these concepts leads to a better understanding of how a photon’s quantum information can be manipulated and controlled.

Studying the dispersion of a two-photon field has lead directly to a deeper understanding of not only physical processes, but also the fundamental properties that reign over quantum communications. These results will further contribute to the creation of more efficient and reliable quantum communication systems, which is of fundamental importance in areas such as quantum cryptography, quantum teleportation, and quantum computing.

## Figures and Tables

**Figure 1 entropy-27-00071-f001:**
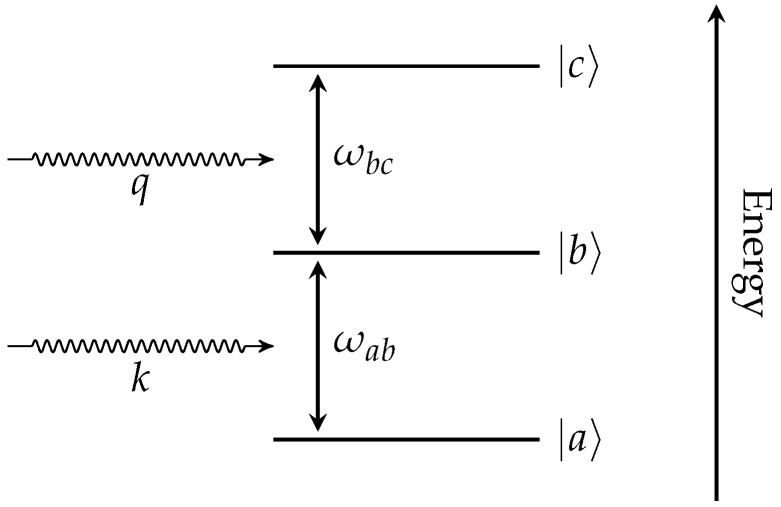
Depiction of the electronic energy structure of a three-level ladder atom interacting with two photons, *k* and *q*.

**Figure 2 entropy-27-00071-f002:**
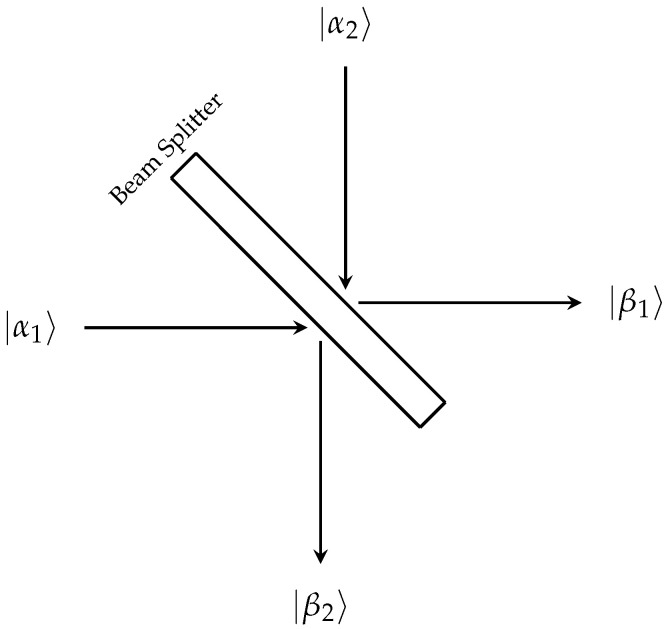
Depiction of the quantum four-port beam splitter. The beam splitter takes in two input states, $|\alpha_{1}\rangle$ and $|\alpha_{2}\rangle$, and produces two output states $|\beta_{1}\rangle$ and $|\beta_{2}\rangle$.

**Figure 3 entropy-27-00071-f003:**
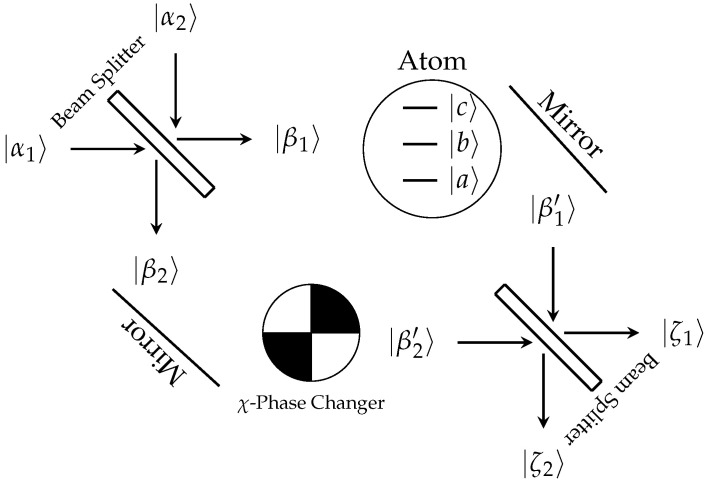
Depiction of the Mach–Zehnder interferometer. In the upper arm resides a three-level atom, while in the lower arm is a variable phase changer.

**Figure 4 entropy-27-00071-f004:**
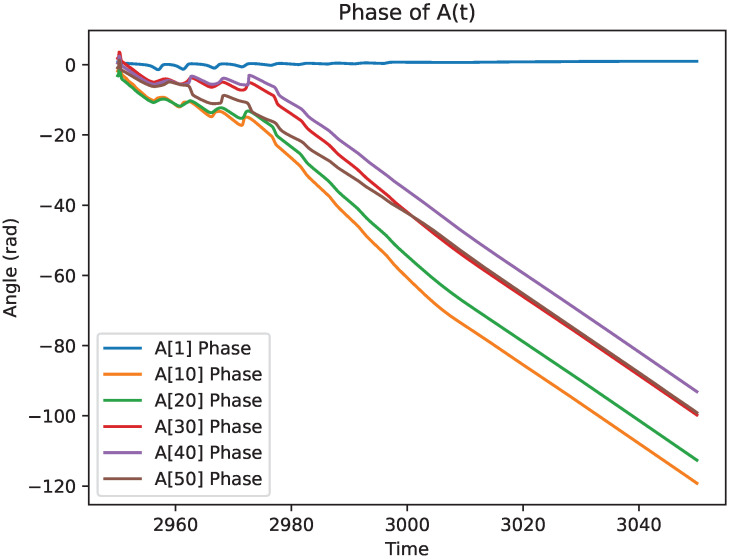
Demonstration of the phase of $A_{k}^{q}\left(t\right)$ for various modes of the k-photon. The *q*-photon was chosen to be in the q=1 mode.

**Figure 5 entropy-27-00071-f005:**
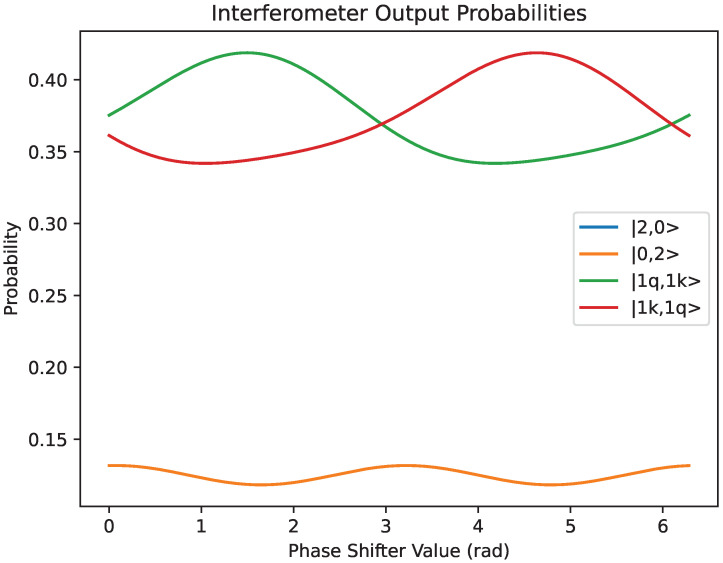
Output configuration probabilities for the Mach–Zehnder interferometer with a χ-phase changer in the lower arm.

**Figure 6 entropy-27-00071-f006:**
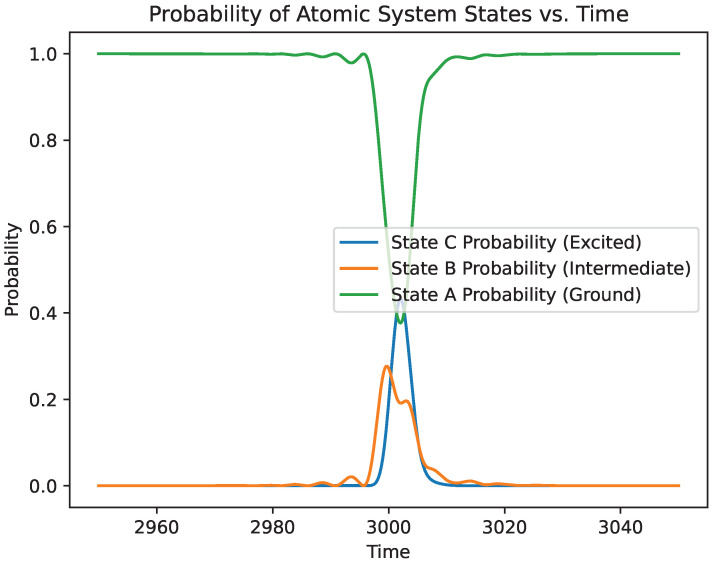
Probability for the electron to be in a given atomic level over the simulation time.

**Figure 7 entropy-27-00071-f007:**
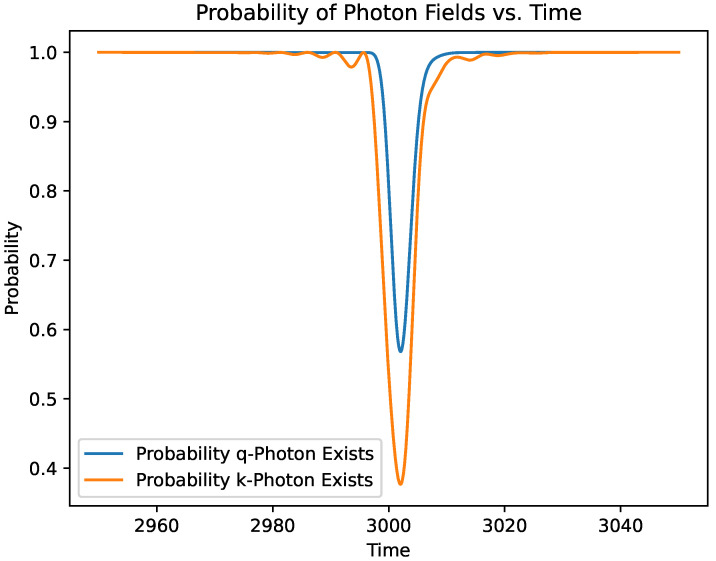
Probability that the *k* and *q* photons exist.

**Figure 8 entropy-27-00071-f008:**
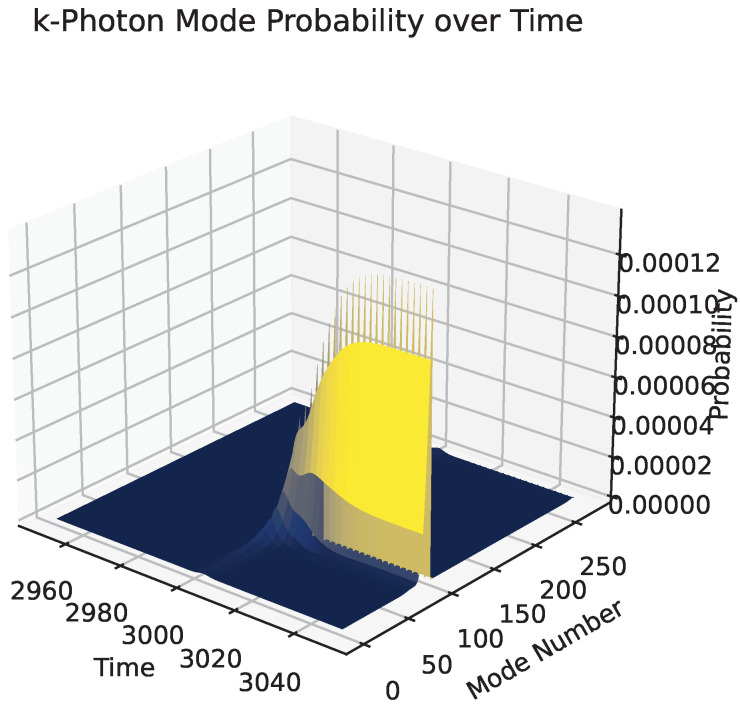
Demonstration of the *k*-photon with equal probability to be in any mode approaching and interacting with the three-level atom. Verification that atomic interaction changes the photon wave packet. The *q* photon is in the q=1 mode.

**Figure 9 entropy-27-00071-f009:**
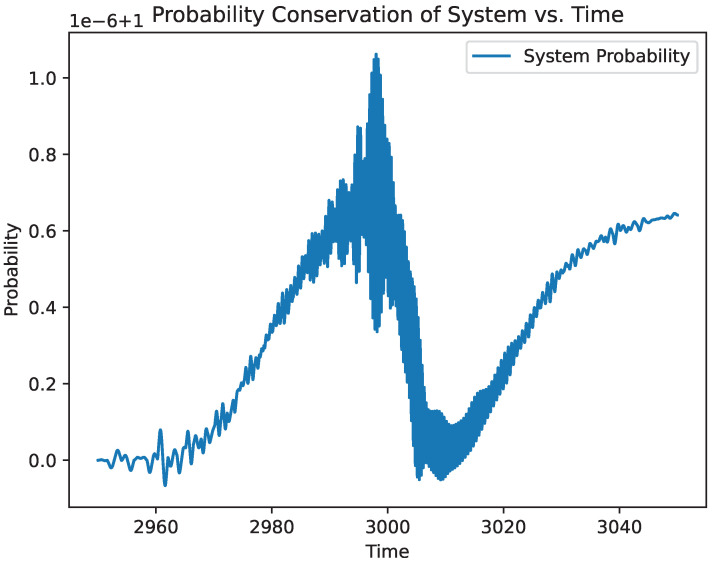
Demonstration of the conservation of probability with the simulated system.

## Data Availability

The original contributions presented in this study are included in the article. Further inquiries can be directed to the corresponding author(s).

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
