# Peer review of "Refraction of the Two-Photon Multimode Field via a Three-Level Atom"

_entropy, 2025, doi:10.3390/e27010071_

Round 1

Reviewer 1 Report

Comments and Suggestions for Authors

The manuscript presents a compelling study on the refractive index of a three-level atom and analyzes the dynamics of a quantum state involving a three-level atom and photons. The authors assert that this work is crucial for advancing quantum technology and enhancing our understanding of quantum physical phenomena. I find the study both interesting and valuable, particularly in the realms of quantum information theory and quantum physics. However, I have the following comments and suggestions:

1. The manuscript employs a Gaussian wave packet to investigate the system's dynamics. Providing physical justification or experimental motivation for this choice would be beneficial. Highlighting the physical relevance or recent experimental advances that make this choice particularly apt would strengthen the manuscript.

2. Some physical parameters should be detailed more thoroughly. For instance, the frequency and time units require clarification. In Figs. 6 and 7, what are the units of time? Providing more context regarding the units of time would enhance the clarity of these figures.

3.  A clear definition of phase velocity and group velocity should be included to aid the reader’s understanding of Fig. 5. Additionally, further explanation is needed regarding the physical interpretation of the results in Fig. 5. What underlying physical processes account for the observed trends?

4. The authors present the case of a single-mode field interacting with the atom. It would be useful to explain why this specific case was chosen. The phenomenon might differ significantly from those in multi-mode fields. Are there any approximations or simplifications that make this case particularly relevant for the study?

5.  In Eq. (28), the state ∣z⟩ might be a typo. Please verify this carefully to ensure the notation is consistent and correct.

6.  If the light field is considered a noisy environment, the coupling could be related to the field's spectrum. Exploring the impact of different spectrum functions, such as a Lorentzian spectrum, would be intriguing. How would this choice affect the results?  

Author Response

Authors’ Response to Reviews of

Refraction of the Two-Photon Multimode Field via a Three- Level Atom

Trever Harborth & Yuri Rostovtsev

Entropy, 3342171

RC: Reviewers’ Comment,     AR: Authors’ Response,     □ Manuscript Text

1.          Reviewer #1

1.1.          Comment #1

RC: The manuscript employs a Gaussian wave packet to investigate the system’s dynamics. Providing physical justification or experimental motivation for this choice would be beneficial. Highlighting the physical relevance or recent experimental advances that make this choice particularly apt would strengthen the manuscript.

AR: We chose the photon envelope to be Gaussian due to its advantageous mathematical, physical, and practical properties. A Gaussian function is smooth and there are no sharp transitions or discontinuities in the pulse. The Fourier transform of a Gaussian function is also a Gaussian.

Many laser systems naturally produce pulses with Gaussian envelopes, as they arise from the fundamental mode of many optical cavities (e.g., TEM00 mode). Gaussian pulses are also naturally emitted in processes involving spontaneous or stimulated emission, where the gain medium supports such distributions. Many ultrashort pulses are generated by mode-locked lasers, which naturally produce pulses with a Gaussian temporal profile. This results from the specific dynamics and stability conditions within the laser cavity. In optical systems, the effects of dispersion and nonlinearities tend to preserve the Gaussian shape of the pulse envelope.

Although optical pulses may not always be perfectly Gaussian, many real-world pulse shapes (e.g., sech2 or Lorentzian) can be approximated by Gaussian envelopes, especially when simplicity and broad applicability are preferred (see Figure 1).

That is why we have chosen the Gaussian envelopes for our paper because they are widely used in optics because they combine mathematical elegance with physical practicality, providing stable, smooth, and efficient solutions for pulse generation, propagation, and manipulation.

The following text has been added:

Figure 1: Comparison of different shapes of envelopes: exp(−x2), sech2(x),        1

1 + x2

1.2.          Comment #2

RC: Some physical parameters should be detailed more thoroughly. For instance, the frequency and time units require clarification. In Figs. 6 and 7, what are the units of time? Providing more context regarding the units of time would enhance the clarity of these figures.

AR: The beginning of the numerical results (Sec. VII) includes that ℏ = c = 1. The following change has been made to more clearly reflect the the simulation is being ran in natural units. A quick refresh is also added for the reader on such conversions. The following changes have been made:

1.3.          Comment #3

RC: A clear definition of phase velocity and group velocity should be included to aid the reader’s understanding of Fig. 5. Additionally, further explanation is needed regarding the physical interpretation of the results in Fig. 5. What underlying physical processes account for the observed trends?

AR:  We agree with the comments of both referees on the group and phase velocities.

As one can see from Eq.(8) of the paper

ig2

ig2            g2              g2

           q f 2 +   k f 2 k  f˙f  q  f˙ f

q    d      k           2

∆2  d d

B˙ (t) =                                                                          k                    q             B(t)                                                  (1)

g2                g2

ℏ2 +   k  f 2 +   q  f 2

d

2                   2

k                   q

The amplitude B(t) acquires a phase

B(t) ≃ exp

         q  f 2 +      k  f 2

B(t0)                                                    (2)

 −ig2             ig2       )

ℏ2∆q d     ℏ2∆k

The result aligns with the results presented in Ref.[1] and extends them to the two-photon regime. The phase and group velocities are defined as follows:

Vph =

= c + c

            f 2 +          k      f 2

ω              −g2

g2             )

(3)

k              ℏ2∆qωd      ℏ2∆kω0

Vg =

= c + c

        q  f 2 +    f 2

(4)

∂ω           g2                g2        )

q

k

∂k              ℏ2∆2  d      ℏ2∆2

The importance of this extension is that the three-level atom has different phase shift than the two-level atom, and it means that the dispersion of the three-level atom can separate the two-photon states from a single photon state.

The text of the paper has been revised accordingly. The following text has been added:

As one can see from Eq.(8), the amplitude

B(t)

acquires the phase

B(t) ≃ exp

  −ig2 R

dtf 2 +

ig2

dtf 2

)B ,

(5)

q

k

,.,.,.,.,.,.,.,.,.,.,.,.,.,.,.,.,.,.,.,.,.,.,.,.,.,.,.,.,.,.,.,.,.,.,.,.,.,.,.,.,.,.,.,.,.,.,.,.,.,.,.,.,.,.,.,.,.,.,.,.,.,.,.,.,.2,.,.q,.,.,.,.,.,.d,.,.,.,.,.2,.,.k,.,.,.,.,.,.,.,.,.,.0,.,.,.

that aligns with the results [1]and extends them to the two-photon regime. Similarly, for a three-level

  −g

g2

g2

atoms, the phase and group velocities can be defined as follows: V

= ω = c + c

               f 2 +            k       f 2

, (6)

,.,.,.,.,.,.,.,.,.,.,.,.,.,.,.,.,.,.,.,.,.,.,.,.,.,.,.,.,.,.,.,.,.,.,.,.,.,.,.,.,.,.,.,.,.,.,.,.,.,.,.,.p,.h,.,.,.,.k,.,.,.,.,.,.,.,.,.,.ℏ2,.,.,.q  ω,.0,.,.d,.,.,.,.,.2,.,.k,.ω,.0,.,.,.,.,.,.,.

,.,.,.,.,.,.,.,.,.,.,.,.,.,.,.,.,.,.,.,.,.,.,.,.,.,.,.,.,.,.,.,.,.,.,.,.,.,.,.,.,.,.,.,.,.,.,.,.,.,.,.,.,.,.,.,.,.,.,.,.,.,.,.,.,. ,.,.,.,.,.,.,.,.,.,.,.,.,.,.,.,.                                                                                                                                                                                                         )

V = ∂ω = c + c

2

q f 2 +

2

g

f 2

).(7) The importance of this extension is that the three-level

atom has different phase shift than the two-level atom, and it means that the dispersion of the three-level

,.,.g,.,.,.,.,.k,.,.,.,.,.,.,.,.,.,.2,.,.2,.,.d,.,.,.,.,.2  ∆,.,.2,.,.,.,.,.,.,.,.,.,.,.,.,.,.,.,.,.,.,.,.,.,.,.,.,.,.,.,.,.,.,.,.,.,.,.,.,.,.,.,.,.,.,.,.,.,.,.,.,.,.,.,.,.

q                       k

atom can provide separation the two-photon states from a single-photon state using the Mach-Zender

,.,.,.,.,.,.,.,.,.,.,.,.,.,.,.,.,.,.,.,.,.,.,.,.,.,.,.,.,.,.,.,.,.,.,.,.,.,.,.,.,.,.,.,.,.,.,.,.,.,.,.,.,.,.,.,.,.,.,.,.,.,.,.,.,.,.,.,.,.,.,.,.,.,.,.,.,.,.,.,.,.

interferometer.

,.,.,.,.,.,.,.,.,.,.,.,.,.,.,.,.,.,.,.,.,.,.,.,.,.,.,.,.,.,.,.,.,.,.,.,.,.,.,.,.,.,.,.,.,.,.,.,.,.,.,.,.,.,.,.,.,.,.,.,.,.,.,.,.,.,.,.,.,.,.,.,.,.,.,.,.,.,.,.,.,.

,.,.,.,.,.,.,.,.,.,.,.,.

1.4.          Comment #4

RC: The authors present the case of a single-mode field interacting with the atom. It would be useful to explain why this specific case was chosen. The phenomenon might differ significantly from those in multi-mode fields. Are there any approximations or simplifications that make this case particularly relevant for the study?

AR: The mode function approach involves selecting an appropriate form for the mode. One common choice is a Gaussian mode function, which effectively consists of a superposition of many plane wave modes.

This choice, however, represents an approximation. If the mode interacts with atoms at resonance, the response becomes more complex. In such cases, the mode undergoes a more intricate phase modulation, rather than simply acquiring a uniform phase shift across the entire envelope.

The behavior of the system differs significantly between the resonant and nonresonant conditions, requiring distinct considerations in each case to accurately describe the dynamics.

The following text has been added:

1.5.          Comment #5

RC:     In Eq. (28), the state |z⟩ might be a typo. Please verify this carefully to ensure the notation is consistent and correct.

AR:  |z, 1q, 1k⟩ is indeed a typo in Eq. (28) and should displayed as |a, 1q, 1k⟩. The following update has been made:

1.6.          Comment #6

RC: If the light field is considered a noisy environment, the coupling could be realted to the field’s spectrum. Exploring the impact of different spectrum functions, such as a Lorentzian spectrum, would be intriguing. How would this choice affect the results?

AR: The concept of a noisy environment can be interpreted in several ways. One perspective involves the presence of background radiation, such as thermal radiation or scattered light in the atmosphere. Another key aspect of a noisy environment is the occurrence of "collisions" or perturbations affecting the single atom that interacts with our system.

These effects are particularly significant in the context of long-distance propagation of single photons, which serve as qubits for quantum communication. Accounting for these noise sources is crucial to understanding how information loss occurs during propagation. That can be done (and we plan to do that in the future) to understand the loss of information during the long-distance propagation reported in Ref.[4] of our paper. However, as now, such a study is beyond the scope of the current paper because it requires the change of Master equation to include the interaction with noisy environment. Please, see comment 2.1 for more details.

Reviewer 2 Report

Comments and Suggestions for Authors

This interesting manuscript describes the dispersion of a two-photon multimode field interacting with a single three-level atom. The manuscript presents that the phase shift introduced by a single atom can be detected by using a Mach-Zehnder interferometer. The paper provides a detailed analysis of the fact that the index of refraction can also be understood from the perspective of the interaction of two-photon with a single three-level atom.

The manuscript is written well and the results will likely interest the quantum information, quantum computing and quantum communication community. I recommend the publication of this manuscript in Entropy after the authors address my comments given below.

 (1) The analysis presented in the manuscript doesn’t take into account the interaction of the system with the bath. In this regard, I wonder how the atom’s decay affects the results described in the manuscript. Authors should include a discussion on this aspect.

  (2) From a more technical point of view, how does a single atom get trapped at a particular location in the Mach-Zehnder interferometer? Please provide a discussion on this.

  (3)  While detecting the phase shift in the Mach-Zehnder interferometer, what is the polarization state of two photons entering the interferometer? In a realistic system, certain selection rules should be obeyed for a photon to interact with particular atomic levels. In this regard, along with energy, the polarization state of the photon is important to describe. For instance, why does photon in a k-mode interact between states |a> and |b> and q-mode between |b> and |c>?

(4) In the manuscript, the three-level atom is considered in the ladder configuration. How do the results differ if the atom is in a lambda and V-type configuration? Please discuss it in the paper.

(5)  What is the significance of negative and positive group velocity in Fig. 5? Does it refer to the superluminal and subluminal group velocity of photons in the system under consideration? Please provide clarification.

Author Response

Authors’ Response to Reviews of

Refraction of the Two-Photon Multimode Field via a Three- Level Atom

Trever Harborth & Yuri Rostovtsev

Entropy, 3342171

RC: Reviewers’ Comment,     AR: Authors’ Response,     □ Manuscript Text

2.          Reviewer #2

2.1.          Comment #1

RC: The analysis presented in the manuscript doesn’t take into account the interaction of the system with the bath. In this regard, I wonder how the atom’s decay affect the results described in the manuscript. Authors should include a discussion on this aspect.

AR: Indeed, we agree with the referee of our paper. We have not specifically studied the effects of relaxation of the excited atomic states. Even though our formalism in the paper, see Eq.(26) and Eq.(28) that take into account the effect of the bath of vacuum modes. In our simulations it was not taken into account, because we are interested in the dispersion of the envelop, that the photon acquires the phase. The relaxation of the atomic system can be easily done, and it will introduce some decay of the photon during propagation, every time photon is absorbed, the decay can be out of a single mode of consideration. These processes are more important if we consider the resonant interaction of a photon with atoms. Here we have a far-detuned photon (for the mode function approach to be valid, see also our answer on remark 1.4).

The following has been added to the Conclusions paper for clarification of this remark.

2.2.          Comment #2

RC: From a more technical point of view, how does a single atom get trapped at a particular location in the Mach-Zehnder interferometer? Please provide a discussion on this.

AR: This is a very good point. Figure 3 may be misleading as an experimental system setup. Figure 3 works as a means to explain the theoretical setup. Experimentally, the atom may be contained in its cavity while the upper arm beam is focused towards the atom via additional mirrors. Of course, this changes the balance of the interferometer as addiational phase is gained from the now longer upper arm. The experimentalist would thus need to calibrate the interferometer and χ-Phase Changer for the adjusted arm length. The following has been added to the manuscript for additional clarification

And after Eq.(72) we added the follwoing text:

2.3.          Comment #3

RC: While detecting the phase shift in the Mach-Zehnder interferometer, what is the polarization state of two photons entering the interferometer? In a realistic system, certain selection rules should be obeyed for a photon to interact with particular atomic levels. In this regard, along with energy, the polarization state of the photon is important to describe. For instance, why does photon in a k-mode interact between states |a⟩ and |b⟩ and q-photon mode between |b⟩ and |c⟩?

AR: The polarization state of the for both photons follows the formalism given in Eq. (3) to Eq. (4) on the right hand side of page 2. In the case presented in the manuscript, linearly polarized light with real polarization vectors as the basis were chosen for both analytic simplicity as well as computational efficiency. The following has been added to the end of the introduction for Sec. II to make this more evident.

Selection rules based off of this is briefly discussed on the right hand side of page three directly after Eq. (19).

AR: A basic reasoning for allowing the k-photon to interact with only the lower energy level and the q-photon with the upper energy level was given on the right hand side of page 4 above Eq. (21). This has been expanded on as shown below:

2.4.          Comment #4

RC: In the manuscript, the three-level atom is considered in the ladder configuration. How do the results differ if the atom is in a lambda and V-type configuration? Please discuss it in the paper.

AR: We believe the results of a V-type and Λ-type atom setup would be very interesting as well for comparisons. However, we feel such investigations are beyond the scope of the current paper. The reasoning for this is it would require a different Hamiltonian for investigation, one that includes two electrons and electron repulsion. Furthermore, some of the approximations/assumptions used in the ladder scheme should not be used for these systems. In particular, the V-type atom’s upper energy levels may be close together in energy resulting in both the k-photon and q-photon to have high probabilities of interactions. Please see comment 2.2 for more details.

2.5.          Comment #5

RC: What is the significance of negative and positive group velocity in Fig. 5? Does it refer to the superluminal and subluminal group velocity of photons in the system under consideration? Please provide clarification.

AR:  We agree with the comments of both referees on the group and phase velocities.

As one can see from Eq.(8) of the paper

ig2

ig2            g2              g2

           q f 2 +   k f 2 k  f˙f  q  f˙ f

q    d      k           2

∆2  d d

B˙ (t) =                                                                          k                    q             B(t)                                                  (8)

g2                g2

ℏ2 +   k  f 2 +   q  f 2

d

2                   2

k                   q

The amplitude B(t) acquires a phase

B(t) ≃ exp

         q  f 2 +      k  f 2

B(t0)                                                    (9)

 −ig2             ig2       )

ℏ2∆q d     ℏ2∆k

The result aligns with the results presented in Ref.[1] and extends them to the two-photon regime. The phase and group velocities are defined as follows:

Vph =

= c + c

            f 2 +          k      f 2

ω              −g2

g2             )

(10)

k              ℏ2∆qωd      ℏ2∆kω0

Vg =

= c + c

        q  f 2 +    f 2

(11)

∂ω           g2                g2        )

q

k

∂k              ℏ2∆2  d      ℏ2∆2

The importance of this extension is that the three-level atom has different phase shift than the two-level atom, and it means that the dispersion of the three-level atom can separate the two-photon states from a single photon state.

The text of the paper has been revised accordingly. The following text has been added:

As one can see from Eq.(8), the amplitude

B(t)

acquires the phase

B(t) ≃ exp

  −ig2 R

dtf 2 +

ig2

dtf 2

)B ,

(12)

q

k

,.,.,.,.,.,.,.,.,.,.,.,.,.,.,.,.,.,.,.,.,.,.,.,.,.,.,.,.,.,.,.,.,.,.,.,.,.,.,.,.,.,.,.,.,.,.,.,.,.,.,.,.,.,.,.,.,.,.,.,.,.,.,.,.,.2,.,.q,.,.,.,.,.,.d,.,.,.,.,.2,.,.k,.,.,.,.,.,.,.,.,.,.0,.,.,.,.

that aligns with the results [1]and extends them to the two-photon regime. Similarly, for a three-level

  −g

g2

g2

atoms, the phase and group velocities can be defined as follows: V

= ω = c + c

               f 2 +            k       f 2

, (13)

,.,.,.,.,.,.,.,.,.,.,.,.,.,.,.,.,.,.,.,.,.,.,.,.,.,.,.,.,.,.,.,.,.,.,.,.,.,.,.,.,.,.,.,.,.,.,.,.,.,.,.,.p,.h,.,.,.,.k,.,.,.,.,.,.,.,.,.,.ℏ2,.,.,.q  ω,.0,.,.d,.,.,.,.,.2,.,.k,.ω,.0,.,.,.,.,.,.,.,.

,.,.,.,.,.,.,.,.,.,.,.,.,.,.,.,.,.,.,.,.,.,.,.,.,.,.,.,.,.,.,.,.,.,.,.,.,.,.,.,.,.,.,.,.,.,.,.,.,.,.,.,.,.,.,.,.,.,.,.,.,.,.,.,.,. ,.,.,.,.,.,.,.,.,.,.,.,.,.,.,.,.                                                                                                                                                                                                         )

V = ∂ω = c + c

2

q f 2 +

2

g

f 2

).(14) The importance of this extension is that the three-level

atom has different phase shift than the two-level atom, and it means that the dispersion of the three-level

,.,.g,.,.,.,.,.k,.,.,.,.,.,.,.,.,.,.2,.,.2,.,.d,.,.,.,.,.2  ∆,.,.2,.,.,.,.,.,.,.,.,.,.,.,.,.,.,.,.,.,.,.,.,.,.,.,.,.,.,.,.,.,.,.,.,.,.,.,.,.,.,.,.,.,.,.,.,.,.,.,.,.,.,.,.,.

q                       k

atom can provide separation the two-photon states from a single-photon state using the Mach-Zender

,.,.,.,.,.,.,.,.,.,.,.,.,.,.,.,.,.,.,.,.,.,.,.,.,.,.,.,.,.,.,.,.,.,.,.,.,.,.,.,.,.,.,.,.,.,.,.,.,.,.,.,.,.,.,.,.,.,.,.,.,.,.,.,.,.,.,.,.,.,.,.,.,.,.,.,.,.,.,.,.,.

interferometer.

,.,.,.,.,.,.,.,.,.,.,.,.,.,.,.,.,.,.,.,.,.,.,.,.,.,.,.,.,.,.,.,.,.,.,.,.,.,.,.,.,.,.,.,.,.,.,.,.,.,.,.,.,.,.,.,.,.,.,.,.,.,.,.,.,.,.,.,.,.,.,.,.,.,.,.,.,.,.,.,.,.

,.,.,.,.,.,.,.,.,.,.,.,.

Round 2

Reviewer 2 Report

Comments and Suggestions for Authors

The authors have implemented my suggestions given in previous report. I suggest to accept the paper in its current form.